# Stochastic Lifting for Generating Trajectories of Stochastic Physical Systems

**Jules Berman** [1]   **Tobias Blickhan** [1]   **Benjamin Peherstorfer** [1]

## Abstract

Many stochastic physical systems evolve smoothly over time in the sense that the distribution of states changes regularly across time steps. The transition from current state to the next state can often be modeled as the combination of a smooth map and an explicit source of randomness. Stochastic Lifting exploits this structure by attaching an independent, high-dimensional random label to each state transition in the training data and fitting a transition map from the current state and label to the next state using a standard regression loss. The labels act as auxiliary coordinates that let the model represent multiple plausible next states from similar current states, avoiding collapse to a mean prediction in the finite-sample size regime. At inference, fresh labels are sampled at each time step and the learned map is rolled forward autoregressively, generating diverse trajectories with a single network evaluation per time step.

## 1. Overview

Many problems in computational physics require drawing large numbers of trajectories from stochastic time-dependent systems. A common abstraction is that dynamics induce a transition law $\boldsymbol{X}_{t+1} \sim \rho(\cdot|\boldsymbol{X}_t = \boldsymbol{x}_t)$ conditional on the previous state $\boldsymbol{x}_t$. The learning problem that we study is therefore not point prediction but sampling plausible next states given a current state $\boldsymbol{x}_t$.

A fundamental difficulty is that fitting a map to training data $(\boldsymbol{x}_t, \boldsymbol{x}_{t+1})$ with standard mean-squared-error regression collapses to the conditional expectation $\mathbb{E}[\boldsymbol{X}_{t+1}|\boldsymbol{X}_t = \boldsymbol{x}_t]$ and therefore suppresses the intrinsic variability of the stochastic trajectories. At the other extreme, general-purpose generative models such as diffusion- and flow-based approaches

[1]Courant Institute of Mathematical Sciences, New York University, New York, NY 10012, USA. Correspondence to: Jules Berman <jmb1174@nyu.edu>.

*Proceedings of the 43rd International Conference on Machine Learning*, Seoul, South Korea. PMLR 306, 2026. Copyright 2026 by the author(s).

can represent distributions, but they typically require multi-step sampling at inference: roughly speaking, generating $T$ time steps with an $s$-step sampler requires $T \times s$ neural-network evaluations (Chen et al., 2024; Kohl et al., 2026). Recent work on one-step samplers aims to reduce this cost by distilling multi-step generation into a single evaluation, but such methods are often more difficult to train reliably and have primarily been demonstrated in stationary settings that differ from the trajectory generation setting that we are interested in (Song et al., 2023; Zhou et al., 2025; Geng et al., 2025a; Boffi et al., 2025). We defer a detailed discussion to the literature review in the next section.

**A smoothness premise specific to time-dependent systems and trajectory data** Our approach is based on a premise that is natural for physical systems; and more broadly for many time-dependent dynamical systems: for sufficiently small time steps, the conditional transition law $\rho(\cdot|\boldsymbol{x}_t)$ varies regularly with the state $\boldsymbol{x}_t$, and next states can be interpreted as small perturbations of the current state. This regularity suggests that the transition can be (at least approximately) represented by a smooth map that takes the current state together with an explicit source of randomness and outputs a next state. The key point is the smoothness of the map. We therefore use smoothness as a guiding heuristic: when a learned transition map interpolates the training data and is smooth, then evaluating it with fresh randomness is expected to generate plausible next-state samples that are close to the desired conditional law.

**Stochastic Lifting** Stochastic Lifting augments the training-data pairs $(\boldsymbol{x}_t, \boldsymbol{x}_{t+1})$ with an independently drawn auxiliary stochastic label $\boldsymbol{\xi}_t$. It then trains a map $F : (\boldsymbol{x}_t, \boldsymbol{\xi}_t) \mapsto \boldsymbol{x}_{t+1}$ by minimizing a standard empirical regression loss over the labeled pairs. The label acts as an auxiliary coordinate that parametrizes stochasticity: by assigning a distinct random label to each realized transition in the training data, the map $F$ can fit multiple possible next states associated with similar or identical current states, rather than being driven by the regression loss to collapse toward the average. Moreover, the dimension of the label provides a handle on regularity to some extent: increasing the label dimension improves the separability of labeled training-data pairs and, as we show, can reduce the minimal Lipschitz constant among interpolants of the training data. In this way, stochastic labeling supports learning maps that

interpolate the training data and remain smooth, aligning the learned maps with the smoothness premise outlined above.

At inference time, we take the current state, draw a fresh label from the label distribution and then evaluate the learned map to obtain a next-state sample. Repeating this procedure with fresh labels yields multiple plausible next-state samples. Furthermore, auto-regressively rolling out the map with a fresh label at each time step leads to a trajectory, requiring only one neural-network evaluation per time step.

**Regime and scope** An important nuance is that Stochastic Lifting is *not* aiming to yield a meaningful population-limit solution. Indeed, even with the randomly labeled data, the population minimizer collapses to the conditional expectation and ignores the labels. Stochastic Lifting is instead intended for the finite-sample regime, where the labeled data can be fit closely, often up to interpolation. Accordingly, instead of convergence in the classical population limit, we are interested in how close the samples that are generated with Stochastic Lifting are compared to actual samples. Our theory is intended as qualitative finite-sample guidance for this regime, highlighting the role of Lipschitz regularity and label dimension rather than guaranteeing convergence or giving tight generalization bounds.

Empirically, we demonstrate state-of-the-art accuracy for generating trajectories of stochastic time-dependent physical systems compared to, e.g., auto-regressive diffusion models and one-step samplers. At the same time, we stress that our method is not designed for unconditional sample generation from noise (e.g., images from noise), where the corresponding map is often globally complex and generally falls outside the regularity regime we rely on. The transition-learning setting, by contrast, provides the structural regularity that makes Stochastic Lifting effective.

**Summary of contributions** **(1)** Introduce Stochastic Lifting, which assigns stochastic labels to transitions to learn a next-step generator; sampling fresh labels at inference time enables trajectory generation with a single network evaluation per time step.

**(2)** Provide finite-sample Wasserstein bounds showing how interpolation together with Lipschitz regularity can control sampling error, and show that higher-dimensional labels can improve the finite-sample geometry that makes low-Lipschitz interpolation possible.

**(3)** Demonstrate state-of-the-art trajectory generation for stochastic, time-dependent physical systems.

## 2. Setup and Problem Formulation

### 2.1. Setup

Consider a stochastic process $\{\boldsymbol{X}_t\}_t \subset \mathcal{X} = [0,1]^n$ with initial condition $\boldsymbol{X}_0 \sim \rho_0$. We denote a state (realization) of

$\boldsymbol{X}_t$ as $\boldsymbol{x}_t$ and often refer to it as state. The samples evolve via a conditional distribution

$$\boldsymbol{X}_{t+1}|\boldsymbol{X}_t = \boldsymbol{x}_t \quad \sim \quad \rho(\,\cdot\,|\boldsymbol{x}_t)\,. \tag{1}$$

To ease exposition, we focus on transitions that condition on the previous state $\boldsymbol{X}_t = \boldsymbol{x}_t$ only, but all of the following extends to transitions that condition on a history of more than one previous state. By the law of total probability, we define the time marginals as $\rho_{t+1} = \int_{\mathcal{X}} \rho(\cdot|\boldsymbol{x}_t)\rho_t(\boldsymbol{x}_t)\,\mathrm{d}\boldsymbol{x}_t$ with the assumption that we can directly sample from $\rho_0$. Notice that the time marginals $\rho_t$ can change with time while the conditional distribution $\rho(\cdot|\boldsymbol{x}_t)$ is independent of time and only depends on the previous state $\boldsymbol{x}_t$. The coupling between $\boldsymbol{x}_t, \boldsymbol{x}_{t+1}$ is described by the density $\pi_t$ over $\mathbb{R}^n \times \mathbb{R}^n$ as $\pi_t(\boldsymbol{x}_t, \boldsymbol{x}_{t+1}) = \rho(\boldsymbol{x}_{t+1}|\boldsymbol{x}_t)\rho_t(\boldsymbol{x}_t)$.

Our data consists of pairs of states

$$\mathcal{D} = \bigcup_{i=1}^M \bigcup_{t=0}^{T-1} \left\{(\boldsymbol{x}_t^i, \boldsymbol{x}_{t+1}^i)\right\} \tag{2}$$

with $(\boldsymbol{x}_t^i, \boldsymbol{x}_{t+1}^i) \sim \pi_t$ for all $i = 1, \ldots, M$. In this sense, we assume that we have available paired data. The goal is to learn a map $F$ that mimics the transition (1). Once we have such a map, we can roll it out autoregressively to rapidly predict new trajectories.

### 2.2. Challenges of learning one-step transition functions of stochastic processes

**Stochastic dynamics lead to one-to-many maps** There cannot exist a function $f : \mathcal{X} \to \mathcal{X}, \boldsymbol{x}_t \mapsto \boldsymbol{x}_{t+1}$ that describes the transition (1) of the stochastic process $\{\boldsymbol{X}_t\}_t$ over more than one time step. The reason is that, when $\rho(\cdot|\boldsymbol{x}_t)$ does not collapse to a single point, the transition from $\boldsymbol{x}_t$ to $\boldsymbol{X}_{t+1}|\boldsymbol{X}_t = \boldsymbol{x}_t$ can be interpreted as a one-to-many map, see Figure 1a. There are arbitrarily many different realizations $\boldsymbol{x}_{t+1}^j, j = 1, 2, 3, \ldots$ of $\boldsymbol{X}_{t+1}|\boldsymbol{X}_t = \boldsymbol{x}_t$ at time $t + 1$ and a function $f$ can map from $\boldsymbol{x}_t$ to only one of them. Even if we consider two different but very close $\boldsymbol{x}_t$ and $\boldsymbol{x}_t'$, a similar issue can arise when the corresponding $\boldsymbol{x}_{t+1}$ and $\boldsymbol{x}_{t+1}'$ are far apart. One can interpret this as a poorly conditioned regression problem, which requires a regression function (and a parametrization that can represent it) that is far from smooth. In particular, the Lipschitz constant of $f$ grows to infeasible values in this setting.

One interpretation of the one-to-many map perspective is that the transition from $\boldsymbol{x}_t$ to a realization of $\boldsymbol{X}_{t+1}|\boldsymbol{X}_t = \boldsymbol{x}_t$ depends on additional information that is not present in $\boldsymbol{x}_t$. For example, let us consider a transition (1) stemming from a discretized stochastic differential equation $\boldsymbol{X}_{t+1} = \boldsymbol{X}_t + \mathrm{d}t\,b(\boldsymbol{X}_t) + \sqrt{\mathrm{d}t}\,\sigma \boldsymbol{W}_t$. The transition from $\boldsymbol{X}_t$ to $\boldsymbol{X}_{t+1}$ depends on the realization of $\boldsymbol{X}_t$ but additionally on the realization of the noise $\boldsymbol{W}_t$, which is typically independent of $\boldsymbol{X}_t$. Thus, $\boldsymbol{x}_t$ alone is insufficient to describe $\boldsymbol{X}_{t+1}|\boldsymbol{X}_t = \boldsymbol{x}_t$.

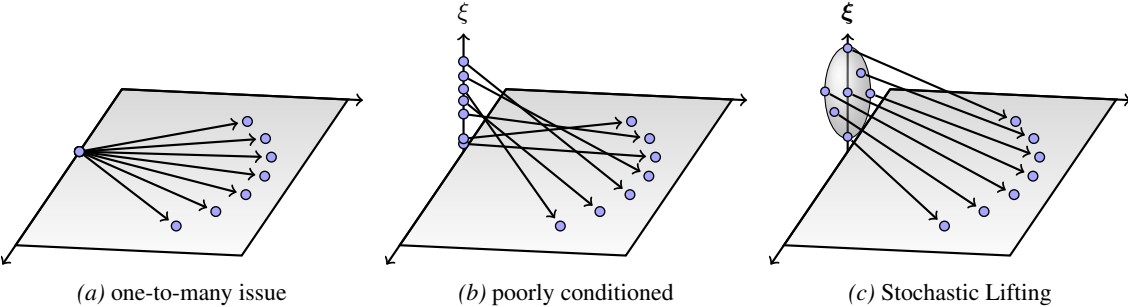

*Figure 1.* **(a)** For data from a stochastic process, (almost) identical states can have different next states, which prevents learning a transition map from just trajectories. **(b)** Labeling data points separates them but low-dimensional labels can still lead to poorly conditioned regression problems. **(c)** Stochastic Lifting uses high-dimensional labels, which enables fitting a smooth regression function.

**Fitting a function to stochastic dynamics collapses to the mean dynamics** Simply fitting a function $f : \mathcal{X} \to \mathcal{X}, \boldsymbol{x}_t \mapsto \boldsymbol{x}_{t+1}$ to data (2) from a stochastic process can capture only mean-like behavior. In fact, training $f$ on data (2) with the mean-squared error loss collapses to the conditional expectation $\boldsymbol{x}_t \mapsto \mathbb{E}[\boldsymbol{X}_{t+1} | \boldsymbol{X}_t = \boldsymbol{x}_t]$ in the limit $M \to \infty$.

If the stochasticity in the data (2) is due to noise and perturbations, then often the goal is recovering the mean behavior, in which case fitting time integrators with the mean-squared loss is reasonable; as in operator learning (Li et al., 2021; Lu et al., 2021; Kovachki et al., 2021). However, our goal is generating new sample trajectories where the randomness is not only injected at the initial condition sampled from $\rho_0$ but also during the roll out, as described by the transition (1) and reflected in the data (2) (recall the example with the stochastic differential equation in the previous paragraph). Thus, capturing mean-like behavior only by applying operator learning and other deterministic transition modeling methods and relying on input randomness alone is insufficient. In particular, we will also consider stochastic processes that always start with the same initial condition $\rho_0 = \delta_{\boldsymbol{x}_0}$ (see wave and multi-phase flow in Section 5).

### 2.3. Literature review

*Multi- and single-step generative models* There is a range of multi-step methods for generative modeling, which build on diffusion- and flow-based modeling (Sohl-Dickstein et al., 2015; Ho et al., 2020; Hyvärinen, 2005; Song et al., 2019; Song & Ermon, 2019; Song et al., 2021; Albergo & Vanden-Eijnden, 2023; Albergo et al., 2025; Lipman et al., 2023; Liu et al., 2023). In contrast, the focus of our work lies on a one-step method, for which a single neural-network function evaluation per next state is sufficient. Early one-step methods are difficult to train (Goodfellow et al., 2014; Arjovsky et al., 2017) or easily collapse (Kingma & Welling, 2014). There are distillation approaches for constructing one-step models out of multi-step models but these require

having readily available a multi-step model (Yin et al., 2024; Sauer et al., 2024; Zhou et al., 2024; Song et al., 2023; Geng et al., 2025b; Frans et al., 2024). More recently, there are methods that aim to circumvent having to train a multi-step model first (Song et al., 2023; Zhou et al., 2025; Geng et al., 2025a; Boffi et al., 2025) but they focus on static generation tasks such as noise to image and do not explicitly leverage the regularity induced by conditioning on previous states in time-dependent problems that we rely on for generating trajectory data.

Note that neural processes and latent state-space models introduce latent variables to model conditional or sequential uncertainty and infer the latents from data (Krishnan et al., 2017; Garnelo et al., 2018). In contrast, Stochastic Lifting does not infer latents or hidden states but assigns independent random labels to transition pairs of states.

*Autoregressive generative models for trajectory data generation* There is a range of works on autoregressive diffusion models (ARDMs) (Hoogeboom et al., 2022; Albergo et al., 2025; Price et al., 2025) and also flow-based models (Davtyan et al., 2023; Chen et al., 2024) that can generate trajectory data; however, these require multiple steps per next state, which can be expensive. There are also time-space approaches (Ho et al., 2022) with the drawback of having to handle a large time-space tensor. We also mention marginal trajectory matching (Neklyudov et al., 2023; Berman et al., 2024; Blickhan et al., 2025) which also performs only a single neural-network function evaluation per time step but the corresponding models can be challenging to scale to high dimensions. Another line of work imposes constraints on the diffusion trajectories to better align them (Rühling Cachay et al., 2023; Ruhe et al., 2024); these again require multiple steps per next state.

*Lifting* A key aspect of our approach is lifting data into higher dimensions, which is prevalent in machine learning (Cortes & Vapnik, 1995; Rahimi & Recht, 2007) but also in computational science in general (McCormick, 1976; Gu, 2011; Kramer & Willcox, 2019; Qian et al., 2020; McQuar-

rie et al., 2021). We mention neural ordinary differential equations (Dupont et al., 2019; Kidger, 2022) that augment the state space to extend the expressivity of neural ordinary differential equations. With lifting we pursue an analogous goal of making a problem better behaved, namely smoother.

## 3. Stochastic Lifting

### 3.1. Labeling data points to avoid one-to-many issue

We randomly draw labels $\boldsymbol{\xi}_t^i \in \mathbb{R}^d$ from a distribution $\nu$ for all $\boldsymbol{x}_t^i$ in the data set $\mathcal{D}$, which leads to the augmented data set

$$\mathcal{D}_\xi = \bigcup_{i=1}^M \bigcup_{t=0}^{T-1} \left\{ (\boldsymbol{x}_t^i, \boldsymbol{x}_{t+1}^i, \boldsymbol{\xi}_t^i) \right\} . \quad (3)$$

In all of the following, the label distribution $\nu$ will be $\mathcal{N}(0, \boldsymbol{I}_d)$, the standard normal distribution of dimension $d$, which implies that the labels are unique almost surely. We assume that $\nu$ is independent of $t$, but we still denote $\boldsymbol{\xi}_t \sim \nu$ to emphasize that, for a given trajectory $\boldsymbol{x}_0^i, \ldots, \boldsymbol{x}_{T-1}^i$, the label of the state at time $t$ need not be the same as the label of the state at time $t + 1$.

Uniquely labeling the data points overcomes the one-to-many issue: Even if there are $\boldsymbol{x}_t^i = \boldsymbol{x}_t^j$ for $i \neq j$, the corresponding labeled points $(\boldsymbol{x}_t^i, \boldsymbol{\xi}_t^i)$ and $(\boldsymbol{x}_t^j, \boldsymbol{\xi}_t^j)$ are unique; see Figure 1b. Thus, circling back to the challenges described in Section 2.2, because of the unique label for each data point, there now exists a function $F : \mathcal{X} \times \mathbb{R}^d \to \mathcal{X}$ that can interpolate (memorize) the data in the sense

$$F(\boldsymbol{x}_t^i, \boldsymbol{\xi}_t^i) = \boldsymbol{x}_{t+1}^i , \quad (4)$$

for $i = 1, \ldots, M$ and $t = 0, \ldots, T - 1$. Recalling the example from Section 2.2 about the stochastic differential equation, we can interpret $\boldsymbol{\xi}_t^i$ as a surrogate of the realization of noise $\boldsymbol{W}_t$ that enters the dynamics.

### 3.2. Wasserstein-2 bound for smooth interpolator

We now show that if the map $F$ is smooth (regular) in terms of its Lipschitz constant and interpolates the data as in (4), then samples generated with $F$ are close to $\rho_{t+1}$ in the Wasserstein-2 metric. Note that smooth refers to Lipschitz regularity of the discrete-time transition map $F$ at a fixed time step on the lifted input space, not to differentiability of sample paths of a continuous-time stochastic process that is possibly underlying the transition law of interest.

**Proposition 3.1.** *Let $F$ interpolate the training data $D_\xi$ as in (4). At any time $t$, consider $\tilde{M}$ test samples $\mathcal{D}_\xi^{test} = \{(\tilde{\boldsymbol{x}}_t^i, \tilde{\boldsymbol{x}}_{t+1}^i, \tilde{\boldsymbol{\xi}}_t^i)\}_{i=1}^{\tilde{M}}$ sampled from $\pi_t \otimes \nu$. Evaluate now the map $F$ to obtain the generated samples $\hat{\boldsymbol{x}}_{t+1}^i = F(\tilde{\boldsymbol{x}}_t^i, \tilde{\boldsymbol{\xi}}_t^i)$ for $i = 1, \ldots, \tilde{M}$. Then,*

$$\mathbb{E}_{\mathcal{D}_\xi, \mathcal{D}_\xi^{test}} \left[ W_2(\hat{\rho}_{t+1}, \tilde{\rho}_{t+1})^2 \right]$$
$$\leq C(1 + L_F^2) \min(M, \tilde{M})^{-2/\alpha} , \quad (5)$$

where $\hat{\rho}_{t+1}$ and $\tilde{\rho}_{t+1}$ are the empirical measures corresponding to the generated samples $\{\hat{\boldsymbol{x}}_{t+1}^i\}_{i=1}^{\tilde{M}}$ and the test samples $\{\tilde{\boldsymbol{x}}_{t+1}^i\}_{i=1}^{\tilde{M}}$. The constant $L_F$ is a deterministic uniform Lipschitz bound on $F$ (see Appendix A.2), $\alpha > 0$ controls the empirical Wasserstein convergence rates of the paired input data $(\boldsymbol{x}_t, \boldsymbol{\xi}_t)$ and of the next-state data $\boldsymbol{x}_{t+1}$, and $C > 0$ is an explicit constant depending only on $\alpha$ and the corresponding empirical Wasserstein constants.

A proof can be found in Appendix A.2. The bound (5) critically depends on the smoothness of $F$ in terms of the Lipschitz constant $L_F$. While simply interpolating independently drawn samples from two distributions can lead to regression functions with high Lipschitz constants in low dimensions (see Figure 1b), we will discuss in the following that labels of high dimension can help to reduce $L_F$.

Furthermore, strong conditioning induced by the previous state can be helpful in regimes where the finite-step transition can be written as the current state plus a regular update. More precisely, at a fixed time discretization we may have $F(\boldsymbol{x}, \boldsymbol{\xi}) = \boldsymbol{x} + \Delta_{\mathrm{dt}}(\boldsymbol{x}, \boldsymbol{\xi})$ with $\Delta_{\mathrm{dt}}$ Lipschitz on the relevant data domain. In drift-dominated settings this update may scale like $\mathrm{dt}$; for Brownian or diffusion-dominated increments it may instead scale like $\sqrt{\mathrm{dt}}$. The results here do not require pathwise differentiability of continuous-time trajectories and do not analyze the limit $\mathrm{dt} \to 0$. The following corollary isolates the special case $\Delta_{\mathrm{dt}} = \mathrm{dt}R$, where the time-step scaling enters in front of the Lipschitz constant and helps alleviate non-smoothness.

**Corollary 3.2.** *If $F(\boldsymbol{x}, \boldsymbol{\xi}) = \boldsymbol{x} + \mathrm{dt}R(\boldsymbol{x}, \boldsymbol{\xi})$, then (5) can be written as*

$$\mathbb{E}_{\mathcal{D}_\xi, \mathcal{D}_\xi^{test}} \left[ W_2(\hat{\rho}_{t+1}, \tilde{\rho}_{t+1})^2 \right]$$
$$\leq C(1 + \mathrm{dt}^2 L_R^2) \min(M, \tilde{M})^{-2/\alpha} , \quad (6)$$

where $C$ and $\alpha$ are as in Proposition 3.1 and $L_R$ is the Lipschitz constant of $R$.

Similar bounds can be derived when assuming more structure on $F$; see Appendix A.2.

For SDE-driven data, our statements apply to the one-step map at the fixed time step used in the data, not to the continuous-time Brownian path or to the limit $\mathrm{dt} \to 0$. For example, a discretized transition may have the form $F(\boldsymbol{x}, \boldsymbol{\xi}) = \boldsymbol{x} + \mathrm{dt}\, b(\boldsymbol{x}) + \sqrt{\mathrm{dt}}\, \sigma\boldsymbol{\xi}$ (see the SDE example in Section 2.2). This finite-step map is within our scope when it is Lipschitz on the data domain, but the relevant Lipschitz constants can depend on $\mathrm{dt}$. In particular, the stochastic term scales like $\sqrt{\mathrm{dt}}$ rather than $\mathrm{dt}$, so Brownian-dominated dynamics do not benefit from the same short-step scaling as drift-dominated dynamics. Stochastic Lifting is therefore expected to work best when $\boldsymbol{x}_t$ and $\boldsymbol{x}_{t+1}$ remain strongly coupled; if the conditional law is nearly independent of $\boldsymbol{x}_t$,

or if pure Brownian noise dominates at the observed time step, then the method falls outside the favorable regime. This coupling requirement is tested directly by the shuffling experiment in Figure 2 and the direct-initial-to-final experiment in Figure 4.

### 3.3. Lifting for smoothness

The key in the previous results was the smoothness of $F$ given by the Lipschitz constant, which we can control to some extent by using high-dimensional labels. The minimal Lipschitz constant of a function $F$ that interpolates the data as in (4) is

$$L(\mathcal{D}_\xi) = \max_{(i,t)\neq(j,s)} \frac{\|\boldsymbol{x}_{t+1}^i - \boldsymbol{x}_{s+1}^j\|_2}{\sqrt{\|\boldsymbol{x}_t^i - \boldsymbol{x}_s^j\|_2^2 + \|\boldsymbol{\xi}_t^i - \boldsymbol{\xi}_s^j\|_2^2}} \,, \quad (7)$$

for $i, j = 1, \ldots, M$ and $t, s = 0, \ldots, T-1$. Note that a function interpolating the data (4) with Lipschitz constant (7) on the data can be extended to $\mathcal{X} \times \mathbb{R}^d$ without increasing the Lipschitz constant (Federer, 1996)[Theorem 2.10.43]. In practice, we rely on standard deep learning regularization techniques (weight decay, normalization layers) to learn a regular interpolant of the data.

**High-dimensional labels decrease the minimal Lipschitz constant**   We draw our labels from a standard normal of dimension $d$. For the sake of the theoretical argument we consider normalized labels with unit norm $\|\boldsymbol{\xi}_t^i\|_2 = \|\boldsymbol{\xi}_s^i\|_2 = 1$, which means that $\frac{1}{2}\|\boldsymbol{\xi}_t^i - \boldsymbol{\xi}_s^j\|_2^2 = 1 - \boldsymbol{\xi}_t^i \cdot \boldsymbol{\xi}_s^j$. That is, the distance between labels is controlled by their inner product. Because the normalized labels are uniformly distributed on the unit sphere $\mathbb{S}^{d-1}$, $\boldsymbol{\xi}_t^i \cdot \boldsymbol{\xi}_s^j$ is of order $1/\sqrt{d}$ with high probability (Vershynin, 2018)[Section 3.3.3 and Theorem 3.3.9]. Hence increasing $d$ widens the separation between labels and reduces the minimal Lipschitz constant of any interpolant as the following proposition shows (see also Figure 1c).

**Proposition 3.3.** *For normalized data and normalized labels, if $M \geq 2$ and $d \geq \max\{c_\delta^2 \ln((T+1)M), 2\}$, then*

$$L(\mathcal{D}_\xi) \leq \frac{\sqrt{n}}{\sqrt{2}} \left(1 + c_\delta \sqrt{\frac{\ln((T+1)M)}{d}}\right),$$

*holds with probability at least $1 - \delta$, $\delta \in (0,1)$, and with constant $c_\delta > 0$ independent of $T, M, d$.*

See Appendix A.1 for a proof. This result is deliberately stated for unit-scale labels: the mechanism is near-orthogonal separation at fixed label norm, not an artificial reduction of the ratio by inflating the label diameter. Thus, we have some control over the minimal Lipschitz constant that any interpolant must have via the label dimension $d$. Note that the regularity of $F$ is further improved by the

closeness between $\boldsymbol{x}_t$ and $\boldsymbol{x}_{t+1}$ (Corollary 3.2). We do not normalize our labels in practice because we expect the components of the label $\boldsymbol{\xi}_t^i$ to be of order one, which is true when $\boldsymbol{\xi}_t^i \sim \mathcal{N}(0, \boldsymbol{I}_d)$. The argument that the minimal $L(\mathcal{D}_\xi)$ can be improved via a high dimension $d$ remains unchanged.

**Orthogonal labels avoid inducing structure**   Traditionally, one would aim to arrange (couple) labels $\boldsymbol{\xi}_t^i$ and targets (next state) $\boldsymbol{x}_{t+1}^i$. We do not rearrange because of costs, and thus we need to ensure that no spurious structure is induced between labels and next states. Choosing the labels close to orthogonal achieves this.

### 3.4. Fitting maps to stochastically lifted data and one-step inference

**Training on lifted data and one-step inference**   We now formulate a regression problem using the labeled data $\mathcal{D}_\xi$. Let us parametrize $F$ as a neural-network function $F_{\boldsymbol{\theta}}$ with weights $\boldsymbol{\theta} \in \mathbb{R}^p$.

We then propose to fit $F_{\boldsymbol{\theta}}$ to the labeled data $\mathcal{D}_\xi$ using a regression-based loss function,

$$\mathcal{L}(\theta, \mathcal{D}_\xi) = \frac{1}{MT} \sum_{i=1}^{M} \sum_{t=0}^{T-1} \|F_{\boldsymbol{\theta}}(\boldsymbol{x}_t^i, \boldsymbol{\xi}_t^i) - \boldsymbol{x}_{t+1}^i\|_2^2. \quad (8)$$

Other regression loss functions can be used. In the physics based experiments we use a L2 error. In the experiments involving natural video we use an LPIPS loss (Zhang et al., 2018); see Appendix F.

To generate a new state at time $t+1$ for a given $\boldsymbol{x}_t$, we draw a new label $\boldsymbol{\xi}_t \sim \nu$ and evaluate the $F_{\boldsymbol{\theta}}(\boldsymbol{x}_t, \boldsymbol{\xi}_t) = \boldsymbol{x}_{t+1}$ to obtain $\boldsymbol{x}_{t+1}$. This is one-step inference because only a single function evaluation is necessary to generate a new sample.

**Lifting avoids collapse onto conditional expectation (mean behavior) of empirical-risk minimizer**   Critically, when taking the infinite data limit $M \to \infty$, then the minimizer of (8) collapses to the conditional expectation $\mathbb{E}[\boldsymbol{X}_{t+1} | \boldsymbol{X}_t = \boldsymbol{x}_t]$, since the labels $\boldsymbol{\xi}_t^i$ are drawn independently from $\boldsymbol{x}_t^i$. This stresses that the map $F$ that we learn with Stochastic Lifting does *not* converge in a distributional sense to the conditional law $\rho(\cdot|\boldsymbol{x}_t)$ in the conventional limit of taking the number of data points to infinity, $M \to \infty$.

However, this collapse is avoidable at finite sample size. We argue that if the label dimension $d$ is allowed to scale with the number of data points $T \times M$, and if the network $F_{\boldsymbol{\theta}}$ is sufficiently expressive, then the minimizer of the *empirical* mean-squared error loss can interpolate the training data as in (4) and does not collapse to the conditional expectation. In particular, for finitely many training data

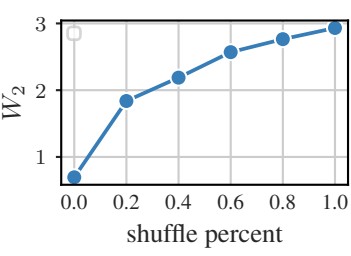 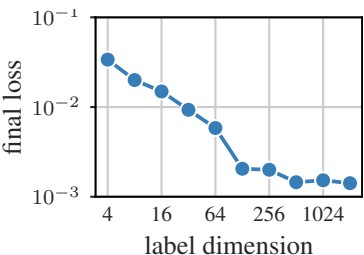 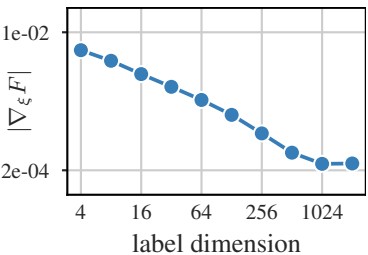

*Figure 2.* Shuffling training pairs $(\boldsymbol{x}_t, \boldsymbol{x}_{t+1})$ breaks the conditional transition structure and degrades Stochastic Lifting's performance, in agreement with the smoothness premise outlined in the Introduction (**left**). Increasing the label dimension helps to push the training into the interpolation regime (**middle**) and induces smoother dependence on the label (**right**), consistent with our discussion.

points $M < \infty$, we can achieve zero training error (interpolation/memorization) by ensuring that $F_\theta$ is expressive enough while at the same time choosing the label dimension $d$ so high that the minimal Lipschitz constant (7) is controlled, in which case a smooth $F_\theta$ can be found. In practice, Adam with weight decay, normalization layers, and standard architecture choices often biases optimization towards learning globally smooth interpolants; see Appendix F. In this regime, we expect $F_\theta$ to behave as discussed in Proposition 3.1, implying that evaluating $F_\theta$ at fresh labels yields outputs that approximately follow the conditional distribution $\rho(\cdot|\boldsymbol{x}_t)$.

As a side remark, we note that in the case of linear least-squares regression, it has been shown that when the ratio $d/(M(T+1))$ between dimension $d$ and number of data points $M(T+1)$ remains fixed in the limit $d, M \to \infty$, then collapse is avoided (Hastie et al., 2022).

## 4. Demonstration on an SDE Example

We demonstrate Stochastic Lifting on a duffing oscillator with random forcing, which is described by an SDE with two-dimensional states; see Appendix D.1. We generate training data, augment each pair $(\boldsymbol{x}_t, \boldsymbol{x}_{t+1})$ with a label, and then train $F_\theta$ represented as a multi-layer perceptron (MLP) with the loss (8). At inference time, we roll out the learned $F_\theta$ as described above from a new sample of our initial distribution and fresh labels in each time step.

**Stochastic Lifting crucially relies on the smoothness premise** In Figure 2 (left), we plot $W_2$ error on test data against the percentage of training pairs $(\boldsymbol{x}_t, \boldsymbol{x}_{t+1})$ that are shuffled prior to training. Concretely, we choose a specified percentage of all training trajectories and, at each time step, independently permute the corresponding $\boldsymbol{x}_{t+1}$ values, thereby breaking the pairing between $\boldsymbol{x}_t$ and $\boldsymbol{x}_{t+1}$. As this shuffle percentage increases, we progressively move away from the true transition coupling toward a product coupling between the time marginals $\rho_t$, in which $\boldsymbol{x}_t$ and $\boldsymbol{x}_{t+1}$ are nearly independent for the shuffled pairs. We observe that increasing the shuffling percentage leads to a larger

$W_2$ error, indicating that Stochastic Lifting indeed critically depends on the smoothness premise in the sense that the conditional transition law $\rho(\cdot|\boldsymbol{x}_t)$ varies smoothly with $\boldsymbol{x}_t$ and next states can be interpreted as small perturbations of the current state.

This experiment directly probes the coupling condition discussed after Corollary 3.2. When the training pairs come from a finite-step transition, the update is typically $\boldsymbol{x}_{t+1} = \boldsymbol{x}_t + \Delta_{\mathrm{dt}}(\boldsymbol{x}_t, \boldsymbol{\xi})$, with $\Delta_{\mathrm{dt}}$ regular on the relevant data domain and, in favorable regimes, small in the time step. Shuffling destroys this structure so that the next state is increasingly drawn as if it were an independent sample from the marginal distribution rather than a sample from the conditional law. The degradation in $W_2$ under shuffling is therefore consistent with the prediction that Stochastic Lifting is most effective in the strongly coupled regime. This also highlights that for SDE-driven data, refining the time step does not make the problem easier indefinitely, because drift increments scale like $\mathrm{d}t$ whereas Brownian increments scale like $\sqrt{\mathrm{d}t}$, so at sufficiently small $\mathrm{d}t$ the stochastic component can dominate the regular deterministic update.

**Higher label dimension enables interpolation and promotes smoothness** In Figure 2 (middle), we plot the final training loss versus label dimension. As the label dimension increases, the loss drops to near zero, indicating that higher-dimensional labels enable the network to interpolate the training data. In Figure 2 (right), we plot the norm of the network gradient against the label dimension. The gradient norm decreases as the label dimension increases, which indicates that the network becomes smoother in the label, supporting our discussion that higher-dimensional labels can promote smoothness.

## 5. Numerical Experiments

**Setup** For all of the following experiments we parametrize our map $F_\theta$ via UNet architectures (Ronneberger et al., 2015) but our method works with other standard diffusion backbone (Peebles & Xie, 2023; Bao et al., 2023; Hoogeboom et al., 2023). We repurpose the diffusion-time input

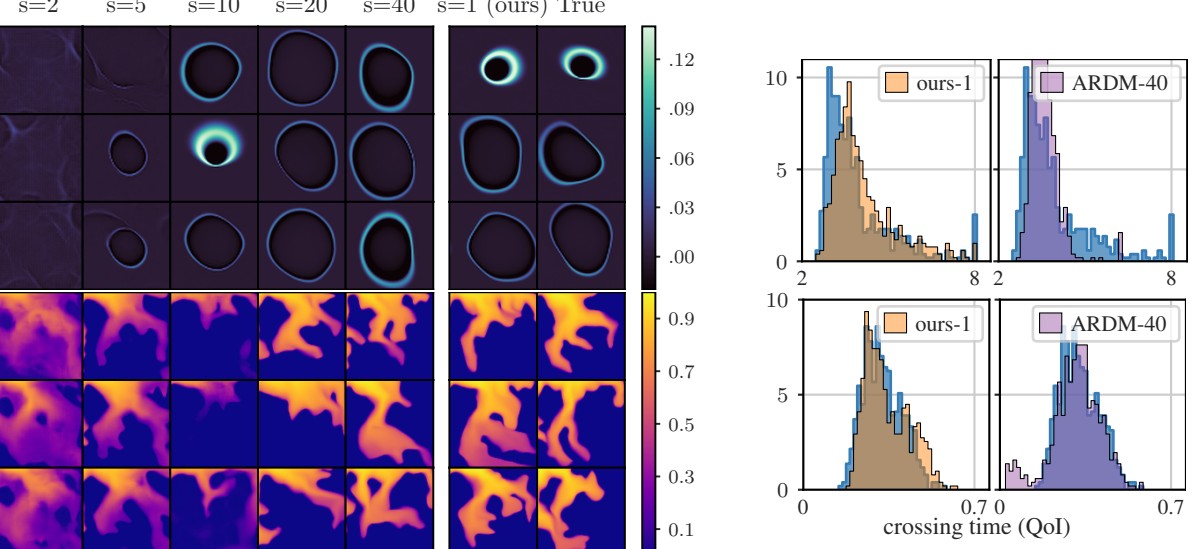

*Figure 3.* **Left**: Stochastic Lifting generates diverse plausible trajectories in just one step per next state (time step), where ARDMs only begin to generate plausible trajectories when using 20 diffusion steps. **Right**: We compute the crossing time for each trajectory (see appendix D.5) and plot the distribution across 512 generated trajectories vs the ground truth. Stochastic Lifting accurately approximates the ground truth data in distribution, even outperforming ARDM using 40 diffusion steps.

to condition on our label $\boldsymbol{\xi}_t$; see Appendix E for details. For all datasets we provide uncurated samples from our model in Appendix G.

We stress that we can use Stochastic Lifting directly on the state space (e.g., pixels of frames) and so avoid having to rely on a latent embedding via pre-trained encoder and decoder networks as many other generative modeling methods (Rombach et al., 2022). By avoiding the latent embedding we ease the significant engineering complexity and hyper-parameter tuning that comes with latent embeddings.

**With just one step per next state, Stochastic Lifting achieves comparable accuracy to multi-step models on physics problems** We consider two physics problems: first, a traveling wave through a spatially varying random medium, which is motivated by seismic wave propagation (Sato et al., 2012); see Appendix D.2. Second, an incompressible two-phase flow in random porous media, which is motivated by petroleum engineering (oil/water) and models of groundwater flow; see Appendix D.3. Importantly, for both wave and flow problems, the initial condition is deterministic and fixed, thus deterministic approaches such as neural operators cannot capture the stochasticity induced by the random media and permeability fields during the autoregressive roll out; see Section 2.2.

We compare our Stochastic Lifting to an autoregressive diffusion model (ARDM) trained on the same dataset. We use the implementation given in (Kohl et al., 2026), also built around a UNet backbone and comparable parameter count as we use for Stochastic Lifting. We vary the number

of diffusion steps $s$ and compare the accuracy to Stochastic Lifting that uses only a single step ($s = 1$).

For the wave and flow problems, ARDM requires 40 steps to generate accurate samples; see Figure 3 (left). In contrast, Stochastic Lifting generates accurate samples in one step. Figure 7–8 in the appendix show nearest-neighbor trajectories and generated states and so provide evidence that Stochastic Lifting is indeed generating new samples instead of simply retrieving memorized ones.

To quantitatively compare the generated samples, we consider the crossing time as a physical quantity of interest, which is the time at which the wave hits the boundary and the saturation reaches the right-bottom corner of the domain, respectively; see Appendix D.5. We plot the distribution of the crossing time of 512 generated trajectories. Stochastic Lifting accurately matches the ground truth data in distribution; see Figure 3 (right). Notice that Stochastic Lifting approximates well the non-Gaussian behavior of the crossing time in the wave example (heavy tail), whereas ARDM fails to capture it even with $s = 40$ steps. Table 1 shows the Wasserstein-2 distance between the distribution of the crossing time (WCT) obtained with Stochastic Lifting and ARDM over various steps. Table 1 also plots the Wasserstein-2 distance (WIM) for another physics quantity of interest, the integrated mass; see Appendix D.5.

**Stochastic Lifting is robust to label dimension, once sufficiently high to allow interpolation** In Figure 4 (left) we plot the final L2 training loss achieved after optimization. For low label dimensions, the neural network cannot

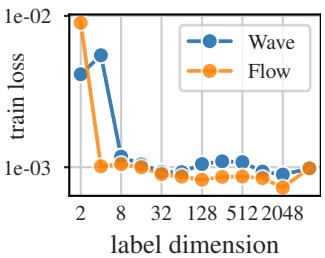 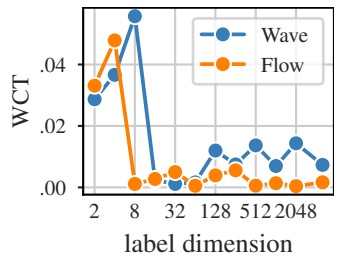 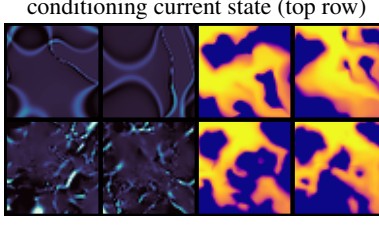

conditioning current state (top row)

direct map from
initial condition alone (bottom row)

*Figure 4.* **Left**: Choosing a higher label dimension $d$ allows the neural network to interpolate the data, resulting in one order-of-magnitude lower L2 final training loss. **Middle**: Stochastic Lifting is robust to the choice of the label dimension as long as it is sufficiently high for interpolation. **Right**: Stochastic Lifting critically depends on the conditioning on the current state to generate a high-quality next state. If we attempt to jump to the last state directly, the method breaks down.

*Table 1.* Wasserstein distance between the distribution of 512 samples from the true model vs. the generative model measured on various quantities of interest; see Appendix D.5.

| | **Wave** | | **Flow** | |
|---|---|---|---|---|
| | **WCT↓** | **WIM↓** | **WCT↓** | **WIM↓** |
| ARDM $s=2$ | 2.49e−2 | 9.11e−5 | 1.18e−1 | 3.46e−2 |
| ARDM $s=5$ | 2.13e−2 | 1.04e−4 | 3.39e−2 | 4.58e−2 |
| ARDM $s=10$ | 1.04e−2 | 6.00e−5 | 9.13e−2 | 1.29e−1 |
| ARDM $s=20$ | 7.28e−3 | 1.06e−4 | 3.25e−2 | 1.27e−3 |
| ARDM $s=40$ | 1.11e−2 | 9.85e−5 | 1.86e−3 | 6.00e−4 |
| **SL (ours)** $s=1$ | 4.53e−3 | 2.06e−5 | 9.47e−4 | 6.08e−4 |

*Table 2.* Stochastic Lifting outperforms all other one-step methods on the BAIR video-generation dataset.

| **One-step methods** ($s = 1$) | |
|---|---|
| SV2P — (Babaeizadeh et al., 2018) | 262.5 |
| SAVP — (Lee et al., 2018) | 109.8 |
| DVD-GAN — (Clark et al., 2019) | 109.8 |
| TrIVD-GAN-FP — (Luc et al., 2020) | 103.3 |
| FitVid — (Babaeizadeh et al., 2021) | 93.6 |
| NUWA — (Wu et al., 2022) | 86.9 |
| **Stochastic Lifting (ours)** | **69.0** |

| **Multi-step methods** | | |
|---|---|---|
| RIVER — (Davtyan et al., 2023) | $s=100$ | 106.1 |
| MCVD — (Voleti et al., 2022) | $s=1000$ | 98.8 |
| RaMViD — (Höppe et al., 2022) | $s=750$ | 84.2 |
| Video Diffusion — (Ho et al., 2022) | $s=16$ | 66.9 |
| Rolling Diffusion — (Ruhe et al., 2024) | $s=32$ | 59.6 |

interpolate the data, resulting in high training loss. Figure 4 (middle) shows that for a sufficiently high label dimension (in particular once interpolation is achieved), the model produces an accurate approximation in the WCT metric (see Table 1). These results align well with Proposition 3.1. At the same time, the approach is fairly robust to the label dimension once it is sufficiently high because architectures such as UNet and the corresponding modulation (see Appendix E) can compress the labels if necessary.

**Stochastic Lifting critically depends on conditioning of current state** In Figure 4 (right, top) we show the final state of the rollout from our model, which is accurate. In the bottom row, we show what happens when we train a Stochastic Lifting model which maps directly from the initial condition $x_0$ to the final state $x_{T−1}$ (i.e. without sequential rollout). In this case, Stochastic Lifting breaks, producing near noise in the wave problem and non-physical, disconnected flows in the flow problem. This study provides further evidence that Stochastic Lifting crucially depends on the "closeness" between the distributions corresponding to the current and next state, which aligns well with Corollary 3.2.

**Stochastic Lifting scales to long-time rollouts (CLEVRER)** The CLEVRER dataset (Yi et al., 2020) is a synthetic dataset designed specifically for video prediction and reasoning tasks. The videos capture diverse objects moving at high speeds from off-screen and colliding with each other. There is inherent stochasticity in when and what objects enter the frame. The training videos are 16 frames long. For testing, we evaluate our model starting with unseen initial frames and then roll out for 500 frames (states), well over one order of magnitude longer than the training videos (compare to 14 frames in (Chen et al., 2024), 120 frames in (Davtyan et al., 2023), 64 frames in (Mei & Patel, 2023)). The long rollout results in objects continuing to enter from off-screen and accumulating in clusters as they collide. Importantly, Stochastic Lifting is able to stably generate the interactions and accumulation of objects, even though such accumulations do not occur in the training set.

**Stochastic Lifting outperforms other one-step methods on BAIR** We apply Stochastic Lifting to the Berkeley AI

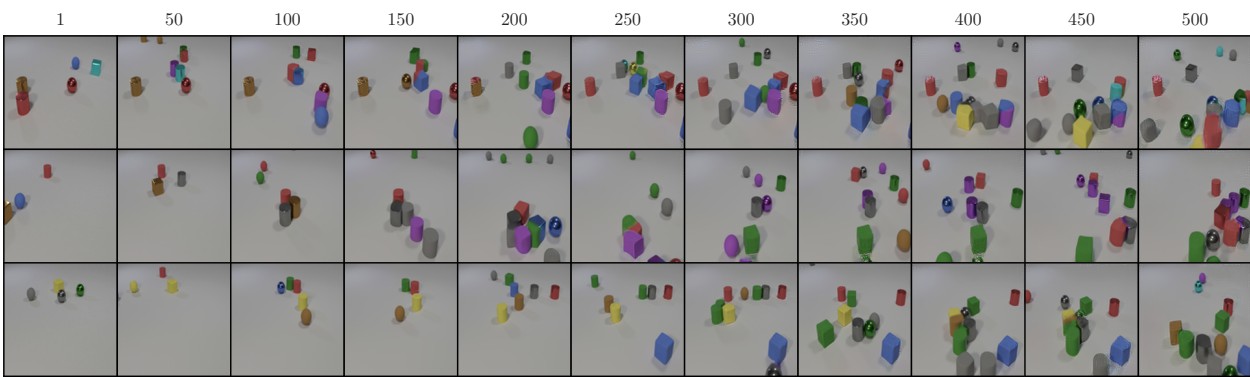

*Figure 5.* We generate 500 frames of a $128 \times 128$ color video in 0.28 seconds on an H100 GPU. The rollout is stable despite being trained on video with only 16 frames. In each trajectory, many more objects are in frame than ever occur during training.

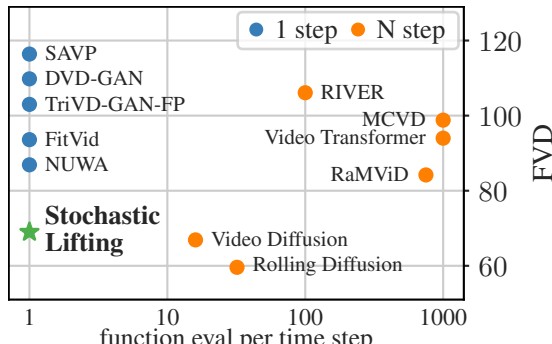

*Figure 6.* On the BAIR video generation benchmark, Stochastic Lifting is state-of-the-art among one-step methods and competitive with diffusion-based methods that use over $10\times$ more compute.

Research (BAIR) robot pushing dataset, which is a standard benchmark (Ebert et al., 2017). It contains roughly $44000$ videos at $64 \times 64$ resolution. The standard prediction task is to generate 15 frames conditioned on one starting frame. We calculate FVD (using the I3D network (Carreira & Zisserman, 2017)) via the standard procedure on BAIR by generating 100 new videos starting from 256 initial frames taken from the test set. FVD is then computed between the generated and test videos. Table 2 shows that Stochastic Lifting achieves the lowest (best) FVD (Unterthiner et al., 2018) over current state-of-the-art one-step generation approaches, and starts closing the gap to multi-step methods (Ruhe et al., 2024).

## 6. Conclusions and Limitations

*Conclusions* We provide evidence with our Stochastic Lifting approach that one can obtain a generative model using a standard empirical regression loss on the data augmented with stochastic labels. The theory provides finite-sample insights by showing that interpolation plus Lipschitz regularity can control Wasserstein sampling error. The provided theoretical results serve three specific roles: First,

they identify smoothness as the key property that makes one-step sampling from an interpolating map accurate (Proposition 3.1), which is critical because interpolation alone could produce arbitrarily poor samples. Second, it provides actionable guidance such as increasing the label dimension improves separability and reduces the minimal Lipschitz constant (Proposition 3.3), telling practitioners that increasing the dimension $d$ can help. Third, it explains why the transition-learning setting is favorable, namely that the small-perturbation structure of consecutive states further reduces the effective Lipschitz constant (Corollary 3.2), clarifying why the method works for trajectory data but not for noise-to-image generation (Figure 4). Empirically, Stochastic Lifting generates stochastic physics trajectories and videos with one network evaluation per time step while matching distributional properties of the data.

*Limitations* (a) We critically build on the strong coupling given by the pair of current and next state in the training data, which means that Stochastic Lifting fails when aiming to generate images from noise. (b) Increasing the label dimension eventually guarantees the existence of a linear interpolant. However, doing so may require proportionally more data; beyond a point, the regression function cannot be trained accurately. (c) Ultimately, the quality of samples created by Stochastic Lifting relies heavily on the fact that the trained network builds a meaningful embedding of the data such that new labels lead to samples within the data manifold. Additional research and theoretical insights are needed to understand when this is the case and when it fails. (d) The current finite-sample theory applies to time marginals only and not to conditional transition laws. Furthermore, tt remains an open question to understand a meaningful population limit of Stochastic Lifting.

## Acknowledgments

The authors have been funded in part by the Air Force Office of Scientific Research (AFOSR), USA, award FA9550-24-1-

0327, and the Defense Advanced Research Projects Agency (DARPA) under Agreement No. HR00112590114, as part of DARPA's Disruptioneering / The Right Space (TRS) effort.

## Impact Statement

This paper presents work whose goal is to advance the field of Machine Learning. There are many potential societal consequences of our work, none which we feel must be specifically highlighted here.

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

# A. Proofs

## A.1. Proof of discrete Lipschitz constant bound

**Proposition A.1.** *Assume that all $\boldsymbol{x}_t^i$ are normalized so that $\boldsymbol{x}_t^i \in [0,1]^n$. Assume further that the labels $\boldsymbol{\xi}_t^i$ are sampled independently and uniformly from the unit sphere $\mathbb{S}^{d-1}$, which corresponds to sampling standard normals and normalizing them afterwards. Let $\mathcal{I} = \{(i,t) \mid i = 1, \dots, M, t = 0, \dots, T-1\}$ to define the discrete Lipschitz constant as*

$$L(D_\xi) = \max_{\substack{(i,t),(j,s) \in \mathcal{I} \\ (i,t) \neq (j,s)}} \frac{\|\boldsymbol{x}_{t+1}^i - \boldsymbol{x}_{s+1}^j\|_2}{\sqrt{\|\boldsymbol{x}_t^i - \boldsymbol{x}_s^j\|_2^2 + \|\boldsymbol{\xi}_t^i - \boldsymbol{\xi}_s^j\|_2^2}} \,.$$

*With*

$$c_\delta = 2 \left( \frac{\sqrt{2}}{\sqrt{c}} + \sqrt{\frac{\ln(2/\delta)}{c \ln(2)}} \right) , \tag{9}$$

*where $c > 0$ is a constant independent of $M, d, T$, if $M \geq 2$ and*

$$d \geq \max\{c_\delta^2 \ln((T+1)M), 2\}, \tag{10}$$

*then with probability at least $1 - \delta$,*

$$L(\mathcal{D}_\xi) \leq \frac{\sqrt{n}}{\sqrt{2}} \left( 1 + c_\delta \sqrt{\frac{\ln((T+1)M)}{d}} \right) . \tag{11}$$

*Proof.* First, notice that

$$\Delta_{\max} = \max_{\substack{(i,t) \neq (j,s) \\ t,s=0,\dots,T}} \|\boldsymbol{x}_t^i - \boldsymbol{x}_s^j\|_2 \leq \sqrt{n}$$

because of the normalization $\boldsymbol{x}_t^i \in [0,1]^n$. Therefore, for every distinct pair $(i,t) \neq (j,s)$, let us consider

$$\frac{\|\boldsymbol{x}_{t+1}^i - \boldsymbol{x}_{s+1}^j\|_2}{\sqrt{\|\boldsymbol{x}_t^i - \boldsymbol{x}_s^j\|_2^2 + \|\boldsymbol{\xi}_t^i - \boldsymbol{\xi}_s^j\|_2^2}} \leq \frac{\sqrt{n}}{\|\boldsymbol{\xi}_t^i - \boldsymbol{\xi}_s^j\|_2} \,,$$

because $\|\boldsymbol{x}_t^i - \boldsymbol{x}_s^j\|_2 \geq 0$ and $\Delta_{\max} \leq \sqrt{n}$. Thus, it suffices to lower bound $\min_{(i,t) \neq (j,s)} \|\boldsymbol{\xi}_t^i - \boldsymbol{\xi}_s^j\|_2$.

Because the labels $\boldsymbol{\xi}_t^i$ are independently and uniformly sampled from the unit sphere $\mathbb{S}^{d-1}$, for the inner product, we have the following concentration inequality that follows from (Vershynin, 2018, Exercise 5.1.12)

$$\mathbb{P}[|\langle \boldsymbol{\xi}_t^i, \boldsymbol{\xi}_s^j \rangle| \geq \epsilon] \leq 2 \exp\left(-cd\epsilon^2\right) , \qquad \epsilon \in [0,1] \tag{12}$$

for a universal constant $c > 0$. (To see (12): Set $f_{\boldsymbol{u}} : \mathbb{S}^{d-1} \to \mathbb{R}, \boldsymbol{z} \mapsto \langle \boldsymbol{z}, \boldsymbol{u} \rangle$ for a fixed $\boldsymbol{u} \in \mathbb{S}^{d-1}$. For any $\boldsymbol{y}, \boldsymbol{z} \in \mathbb{S}^{d-1}$ obtain that $\|f_{\boldsymbol{u}}(\boldsymbol{y}) - f_{\boldsymbol{u}}(\boldsymbol{z})\|_2 = \|\langle \boldsymbol{y} - \boldsymbol{z}, \boldsymbol{u} \rangle\|_2 \leq \|\boldsymbol{y} - \boldsymbol{z}\|_2 \|\boldsymbol{u}\|_2 = \|\boldsymbol{y} - \boldsymbol{z}\|_2$ because $\|\boldsymbol{u}\|_2 = 1$ by definition. Thus the Lipschitz constant of $f_{\boldsymbol{u}}$ is one. Now notice that $\mathbb{E}[f_{\boldsymbol{u}}(\boldsymbol{z})] = 0$ because $\mathbb{E}[\boldsymbol{z}] = 0$ for $\boldsymbol{z}$ uniformly on $\mathbb{S}^{d-1}$. Therefore with (Vershynin, 2018, Exercise 5.1.12) obtain that $\mathbb{P}[|f_{\boldsymbol{u}}(\boldsymbol{z})| \geq \epsilon] \leq 2 \exp(-cd\,\epsilon^2)$ for $\epsilon \in (0,1]$. Taking two independent $\boldsymbol{\xi}_t^i, \boldsymbol{\xi}_s^j$ uniformly on $\mathbb{S}^{d-1}$, we obtain $\mathbb{P}[|\langle \boldsymbol{\xi}_t^i, \boldsymbol{\xi}_s^j \rangle| \geq \epsilon] = \mathbb{E}_{\boldsymbol{\xi}_s^j}[\mathbb{P}[|\langle \boldsymbol{\xi}_t^i, \boldsymbol{\xi}_s^j \rangle| \geq \epsilon | \boldsymbol{\xi}_s^j]] \leq \mathbb{E}_{\boldsymbol{\xi}_s^j}[2 \exp(-cd\,\epsilon^2)] = 2 \exp(-cd\,\epsilon^2)$ because the right-hand side of the bound for $f_{\boldsymbol{u}}$ is independent of $\boldsymbol{\xi}_s^j$.)

Thus, we obtain

$$\mathbb{P}\left[ \sqrt{2(1 - |\langle \boldsymbol{\xi}_t^i, \boldsymbol{\xi}_s^j \rangle|} \leq \sqrt{2(1-\epsilon)} \right] \leq 2 \exp\left(-cd\epsilon^2\right) , \qquad \epsilon \in [0,1] \,,$$

holds. This is useful because $\|\boldsymbol{\xi}_t^i - \boldsymbol{\xi}_s^j\|_2^2 = 2 - 2\langle \boldsymbol{\xi}_t^i, \boldsymbol{\xi}_s^j \rangle$ and thus $\|\boldsymbol{\xi}_t^i - \boldsymbol{\xi}_s^j\|_2 \geq \sqrt{2(1 - |\langle \boldsymbol{\xi}_t^i, \boldsymbol{\xi}_s^j \rangle|)}$. Selecting a $\delta \in (0,1)$ and using a union bound, we obtain

$$\mathbb{P}\left[ \min_{(i,t) \neq (j,s)} \|\boldsymbol{\xi}_t^i - \boldsymbol{\xi}_s^j\|_2 \geq \sqrt{2(1 - \epsilon_\delta)} \right] \geq 1 - \delta \,,$$

for

$$\epsilon_\delta = \sqrt{\frac{1}{cd}\left(\ln\left(\binom{|\mathcal{I}|}{2}\right) + \ln(2/\delta)\right)}.$$

Note that $|\mathcal{I}| = MT \le (T+1)M$ and thus $\binom{|\mathcal{I}|}{2} \le ((T+1)M)^2$ and

$$\ln\left(\binom{|\mathcal{I}|}{2}\right) + \ln(2/\delta) \le 2\ln((T+1)M) + \ln(2/\delta).$$

Using $\sqrt{a+b} \le \sqrt{a} + \sqrt{b}$ and $(T+1)M \ge 2$, we obtain

$$\epsilon_\delta \le \frac{\sqrt{2}}{\sqrt{c}}\sqrt{\frac{\ln((T+1)M)}{d}} + \sqrt{\frac{\ln(2/\delta)}{c\ln(2)}}\sqrt{\frac{\ln((T+1)M)}{d}}.$$

Now set $c_\delta$ to (9) and define

$$\lambda = c_\delta\sqrt{\frac{\ln((T+1)M)}{d}}. \tag{13}$$

Assumption (10) implies $\lambda \le 1$ and with previous estimates we obtain

$$\epsilon_\delta \le \frac{\lambda}{2} \le \frac{1}{2}.$$

Furthermore, $1 - \epsilon_\delta \ge 1 - \lambda/2$. Because $0 \le \lambda \le 1$, we have

$$1 - \frac{\lambda}{2} \ge \frac{1}{(1+\lambda)^2}.$$

Therefore,

$$\sqrt{2(1-\epsilon_\delta)} \ge \frac{\sqrt{2}}{1+\lambda}.$$

Consequently, with probability at least $1 - \delta$,

$$L(D_\xi) \le \frac{\sqrt{n}}{\sqrt{2(1-\epsilon_\delta)}} \le \frac{\sqrt{n}}{\sqrt{2}}(1+\lambda).$$

Substituting (13) for $\lambda$ yields (11). $\qquad\square$

### A.2. Interpolation of training data + smoothness = bound on test data

We recall the following lemma for the sake of self-containedness:

**Lemma A.2.** *For $F : \mathbb{R}^n \to \mathbb{R}^m$ Lipschitz continuous with constant $L_F$ and $\mu, \nu \in \mathcal{P}_2(\mathbb{R}^n)$ (i.e. with finite second moments),*

$$W_2(F_\sharp\mu, F_\sharp\nu) \le L_F W_2(\mu, \nu) \tag{14}$$

*Proof.* Assume $\gamma^* \in \Gamma(\mu, \nu)$ is the optimal coupling between $\mu$ and $\nu$, hence

$$\gamma^*(\cdot, \mathbb{R}^n) = \mu, \ \gamma^*(\mathbb{R}^n, \cdot) = \nu, \text{ and } \int_{\mathbb{R}^n \times \mathbb{R}^n} |\boldsymbol{x} - \boldsymbol{y}|^2 \, d\gamma^*(x, y) = W_2(\mu, \nu)^2. \tag{15}$$

Consider the coupling $\gamma_F := (F \times F)_\sharp\gamma^*$. Take $B \subset \mathbb{R}^m$ measurable. Then,

$$\gamma_F(B \times \mathbb{R}^m) = ((F \times F)_\sharp\gamma^*)(B \times \mathbb{R}^m) = \gamma^*((F \times F)^{-1}(B \times \mathbb{R}^m))$$
$$= \gamma^*(F^{-1}(B) \times F^{-1}(\mathbb{R}^m)) = \gamma^*(F^{-1}(B) \times \mathbb{R}^n) = \mu(F^{-1}(B)) = (F_\sharp\mu)(B). \tag{16}$$

Analogously, $\gamma_F(\mathbb{R}^m \times B) = (F_\sharp \nu)(B)$. Hence $\gamma_F$ is a valid competitor for the optimal transport problem from $F_\sharp \mu$ to $F_\sharp \nu$:

$$W_2(F_\sharp \mu, F_\sharp \nu)^2 = \inf_{\gamma \in \Gamma(F_\sharp \mu, F_\sharp \nu)} \int_{\mathbb{R}^m \times \mathbb{R}^m} |\boldsymbol{x} - \boldsymbol{y}|^2 \, \mathrm{d}\gamma(\boldsymbol{x}, \boldsymbol{y}) \leq \int_{\mathbb{R}^m \times \mathbb{R}^m} |\boldsymbol{x} - \boldsymbol{y}|^2 \, \mathrm{d}\gamma_F(\boldsymbol{x}, \boldsymbol{y})$$

$$= \int_{\mathbb{R}^n \times \mathbb{R}^n} |F(x) - F(y)|^2 \, \mathrm{d}\gamma^*(\boldsymbol{x}, \boldsymbol{y}) \leq L_F^2 \int_{\mathbb{R}^n \times \mathbb{R}^n} |\boldsymbol{x} - \boldsymbol{y}|^2 \, \mathrm{d}\gamma^*(\boldsymbol{x}, \boldsymbol{y}) = L_F^2 W_2(\mu, \nu)^2 \quad (17)$$

Taking the square root gives the claimed result. $\qquad \square$

*Proposition* (3.1). Fix a time $t$. Assume:

**(A1)** The training pairs $(\hat{\boldsymbol{x}}_t^i, \hat{\boldsymbol{x}}_{t+1}^i) \sim \pi_t$ are i.i.d., and independently the training labels $\hat{\boldsymbol{\xi}}_t^i \sim \nu$ are i.i.d. for $i = 1, \ldots, M$. The test data $(\tilde{\boldsymbol{x}}_t^i, \tilde{\boldsymbol{x}}_{t+1}^i, \tilde{\boldsymbol{\xi}}_t^i)_{i=1}^{\tilde{M}}$ is sampled in the same way and is independent of the training data.

**(A2)** All laws involved have finite second moments.

**(A3)** The map $F : \mathcal{X} \times \mathbb{R}^d \to \mathcal{X}$ interpolates the training data, $F(\hat{\boldsymbol{x}}_t^i, \hat{\boldsymbol{\xi}}_t^i) = \hat{\boldsymbol{x}}_{t+1}^i$ for $i = 1, \ldots, M$, and admits a deterministic uniform Lipschitz bound $\mathrm{Lip}(F) \leq L_F$ over all training datasets under consideration.

**(A4)** Denote by $\eta_t$ the joint law of a paired sample $(\boldsymbol{x}_t, \boldsymbol{\xi}_t)$ with $\boldsymbol{x}_t \sim \rho_t$ and $\boldsymbol{\xi}_t \sim \nu$ drawn independently. There exist constants $C_{\mathrm{in}}, C_{\mathrm{out}} > 0$ and exponents $\alpha_{\mathrm{in}}, \alpha_{\mathrm{out}} > 0$ such that the empirical Wasserstein convergence rates

$$\mathbb{E} \, W_2(\eta_t, \hat{\eta}_t^N)^2 \leq C_{\mathrm{in}} N^{-2/\alpha_{\mathrm{in}}}, \quad \mathbb{E} \, W_2(\rho_{t+1}, \hat{\rho}_{t+1}^N)^2 \leq C_{\mathrm{out}} N^{-2/\alpha_{\mathrm{out}}}$$

hold, where $\hat{\eta}_t^N := \frac{1}{N} \sum_{i=1}^N \delta_{(\boldsymbol{x}_t^i, \boldsymbol{\xi}_t^i)}$ denotes the empirical measure of $N$ i.i.d. paired samples from $\eta_t$, and $\hat{\rho}_{t+1}^N$ denotes the empirical measure of $N$ i.i.d. samples from $\rho_{t+1}$.

Let $\hat{\boldsymbol{x}}_{t+1}^i = F(\tilde{\boldsymbol{x}}_t^i, \tilde{\boldsymbol{\xi}}_t^i)$ for $i = 1, \ldots, \tilde{M}$ be the generated test samples. Then,

$$\mathbb{E}_{\mathcal{D}_\xi, \mathcal{D}_\xi^{\mathrm{test}}} \left[ W_2(\hat{\rho}_{t+1}, \tilde{\rho}_{t+1})^2 \right] \leq C(1 + L_F^2) \min(M, \tilde{M})^{-2/\alpha}, \quad (18)$$

where $\hat{\rho}_{t+1}$ and $\tilde{\rho}_{t+1}$ are the empirical measures of the generated samples $\{\hat{\boldsymbol{x}}_{t+1}^i\}_{i=1}^{\tilde{M}}$ and the test samples $\{\tilde{\boldsymbol{x}}_{t+1}^i\}_{i=1}^{\tilde{M}}$, respectively; $\alpha := \max(\alpha_{\mathrm{in}}, \alpha_{\mathrm{out}})$ and $C := 8 \max(C_{\mathrm{in}}, C_{\mathrm{out}})$. When $F(\boldsymbol{x}, \boldsymbol{\xi}) = \boldsymbol{x} + \mathrm{d}t R(\boldsymbol{x}, \boldsymbol{\xi})$ with $R$ Lipschitz on the lifted domain with constant $L_R$, the bound can be refined to

$$\mathbb{E}_{\mathcal{D}_\xi, \mathcal{D}_\xi^{\mathrm{test}}} \left[ W_2(\hat{\rho}_{t+1}, \tilde{\rho}_{t+1})^2 \right] \leq C(1 + \mathrm{d}t^2 L_R^2) \min(M, \tilde{M})^{-2/\alpha}. \quad (19)$$

If instead $F(\boldsymbol{x}, \boldsymbol{\xi}) = \boldsymbol{x} + \mathrm{d}t \, b(\boldsymbol{x}) + \varepsilon \sigma(\boldsymbol{\xi})$ with $b, \sigma$ Lipschitz with constants $L_b, L_\sigma$, then

$$\mathbb{E}_{\mathcal{D}_\xi, \mathcal{D}_\xi^{\mathrm{test}}} \left[ W_2(\hat{\rho}_{t+1}, \tilde{\rho}_{t+1})^2 \right] \leq C(1 + \mathrm{d}t^2 L_b^2 + \varepsilon^2 L_\sigma^2) \min(M, \tilde{M})^{-2/\alpha}. \quad (20)$$

*Remark* A.3. The bound controls the one-step *marginal* of the generated next-state samples at time $t + 1$, drawn under the law $\rho_t$ of the current state. It does not by itself establish conditional accuracy at each fixed current state $\boldsymbol{x}_t$.

*Remark* A.4. Under the compactness (or sub-Gaussian tail) assumptions required for the empirical Wasserstein concentration results of (Weed & Bach, 2019), the empirical-measure terms appearing in the proof also admit high-probability analogues by a union bound; we state the result in expectation for simplicity.

*Proof.* Denote by $\hat{\eta}_t := \frac{1}{M} \sum_{i=1}^M \delta_{(\hat{\boldsymbol{x}}_t^i, \hat{\boldsymbol{\xi}}_t^i)}$ the paired empirical of the training inputs and by $\tilde{\eta}_t := \frac{1}{\tilde{M}} \sum_{i=1}^{\tilde{M}} \delta_{(\tilde{\boldsymbol{x}}_t^i, \tilde{\boldsymbol{\xi}}_t^i)}$ the analogous test paired empirical. By assumption **(A3)**, $F$ interpolates the training data, so the pushforward of the paired training empirical coincides with the training empirical of the next state,

$$F_\sharp \hat{\eta}_t = \frac{1}{M} \sum_{i=1}^M \delta_{F(\hat{\boldsymbol{x}}_t^i, \hat{\boldsymbol{\xi}}_t^i)} = \frac{1}{M} \sum_{i=1}^M \delta_{\hat{\boldsymbol{x}}_{t+1}^i} =: \hat{\rho}_{t+1}^{\mathrm{train}}. \quad (21)$$

The generated empirical from the test inputs satisfies $\hat{\rho}_{t+1} = F_\sharp \tilde{\eta}_t$. By the triangle inequality,

$$W_2(\hat{\rho}_{t+1}, \tilde{\rho}_{t+1}) \leq \underbrace{W_2(F_\sharp \tilde{\eta}_t, F_\sharp \hat{\eta}_t)}_{=:\,(i)} + \underbrace{W_2(\hat{\rho}_{t+1}^{\text{train}}, \tilde{\rho}_{t+1})}_{=:\,(ii)}, \tag{22}$$

where we used $F_\sharp \hat{\eta}_t = \hat{\rho}_{t+1}^{\text{train}}$ in $(ii)$. Squaring and using $(a+b)^2 \leq 2a^2 + 2b^2$,

$$W_2(\hat{\rho}_{t+1}, \tilde{\rho}_{t+1})^2 \leq 2\, W_2(F_\sharp \tilde{\eta}_t, F_\sharp \hat{\eta}_t)^2 + 2\, W_2(\hat{\rho}_{t+1}^{\text{train}}, \tilde{\rho}_{t+1})^2. \tag{23}$$

*Bound on* $(i)$. By Lemma A.2 and assumption **(A3)**,

$$W_2(F_\sharp \tilde{\eta}_t, F_\sharp \hat{\eta}_t)^2 \leq L_F^2\, W_2(\tilde{\eta}_t, \hat{\eta}_t)^2. \tag{24}$$

Since $\hat{\eta}_t$ and $\tilde{\eta}_t$ are i.i.d. paired empirical measures of $\eta_t$ with $M$ and $\tilde{M}$ samples respectively, the triangle inequality and assumption **(A4)** give

$$\mathbb{E}\, W_2(\tilde{\eta}_t, \hat{\eta}_t)^2 \leq 2\,\mathbb{E}\, W_2(\tilde{\eta}_t, \eta_t)^2 + 2\,\mathbb{E}\, W_2(\eta_t, \hat{\eta}_t)^2 \leq 2C_{\text{in}}\big(M^{-2/\alpha_{\text{in}}} + \tilde{M}^{-2/\alpha_{\text{in}}}\big). \tag{25}$$

*Bound on* $(ii)$. By assumption **(A1)**, $\hat{\rho}_{t+1}^{\text{train}}$ and $\tilde{\rho}_{t+1}$ are independent empirical measures of $\rho_{t+1}$ with $M$ and $\tilde{M}$ samples respectively, so by the same argument and assumption **(A4)**,

$$\mathbb{E}\, W_2(\hat{\rho}_{t+1}^{\text{train}}, \tilde{\rho}_{t+1})^2 \leq 2C_{\text{out}}\big(M^{-2/\alpha_{\text{out}}} + \tilde{M}^{-2/\alpha_{\text{out}}}\big). \tag{26}$$

Taking expectation in (23) and combining the two bounds,

$$\mathbb{E}\, W_2(\hat{\rho}_{t+1}, \tilde{\rho}_{t+1})^2 \leq 4L_F^2 C_{\text{in}}\big(M^{-2/\alpha_{\text{in}}} + \tilde{M}^{-2/\alpha_{\text{in}}}\big) + 4C_{\text{out}}\big(M^{-2/\alpha_{\text{out}}} + \tilde{M}^{-2/\alpha_{\text{out}}}\big). \tag{27}$$

Setting $\alpha := \max(\alpha_{\text{in}}, \alpha_{\text{out}})$ and $C := 8\max(C_{\text{in}}, C_{\text{out}})$, and using $M^{-2/\alpha_{\text{in}}} + \tilde{M}^{-2/\alpha_{\text{in}}} \leq 2\min(M, \tilde{M})^{-2/\alpha}$ (and similarly for $\alpha_{\text{out}}$),

$$\mathbb{E}\, W_2(\hat{\rho}_{t+1}, \tilde{\rho}_{t+1})^2 \leq C\,(1 + L_F^2)\,\min(M, \tilde{M})^{-2/\alpha}, \tag{28}$$

which is the claim of Proposition 3.1.

*Proof of the corollary.* Now suppose $F(\boldsymbol{x}, \boldsymbol{\xi}) = \boldsymbol{x} + \mathrm{d}t\, R(\boldsymbol{x}, \boldsymbol{\xi})$ with $R$ Lipschitz with constant $L_R$. Let $\gamma^*$ denote the optimal coupling between $\hat{\eta}_t$ and $\tilde{\eta}_t$ on $(\mathcal{X} \times \mathbb{R}^d)^2$. Then, by Lemma A.2's construction,

$$W_2(F_\sharp \tilde{\eta}_t, F_\sharp \hat{\eta}_t)^2 \leq \int |F(\tilde{\boldsymbol{x}}, \tilde{\boldsymbol{\xi}}) - F(\hat{\boldsymbol{x}}, \hat{\boldsymbol{\xi}})|^2\, \mathrm{d}\gamma^*\big((\hat{\boldsymbol{x}}, \hat{\boldsymbol{\xi}}), (\tilde{\boldsymbol{x}}, \tilde{\boldsymbol{\xi}})\big). \tag{29}$$

Using $F(\boldsymbol{x}, \boldsymbol{\xi}) = \boldsymbol{x} + \mathrm{d}t\, R(\boldsymbol{x}, \boldsymbol{\xi})$ and the inequality $|a + \mathrm{d}t\, b|^2 \leq 2|a|^2 + 2\mathrm{d}t^2 |b|^2$,

$$|F(\tilde{\boldsymbol{x}}, \tilde{\boldsymbol{\xi}}) - F(\hat{\boldsymbol{x}}, \hat{\boldsymbol{\xi}})|^2 = |\tilde{\boldsymbol{x}} - \hat{\boldsymbol{x}} + \mathrm{d}t(R(\tilde{\boldsymbol{x}}, \tilde{\boldsymbol{\xi}}) - R(\hat{\boldsymbol{x}}, \hat{\boldsymbol{\xi}}))|^2 \tag{30}$$

$$\leq 2\,|\tilde{\boldsymbol{x}} - \hat{\boldsymbol{x}}|^2 + 2\mathrm{d}t^2\, L_R^2\, |(\tilde{\boldsymbol{x}}, \tilde{\boldsymbol{\xi}}) - (\hat{\boldsymbol{x}}, \hat{\boldsymbol{\xi}})|^2. \tag{31}$$

We now integrate against $\gamma^*$ and use that the marginal of $|(\tilde{\boldsymbol{x}}, \tilde{\boldsymbol{\xi}}) - (\hat{\boldsymbol{x}}, \hat{\boldsymbol{\xi}})|^2$ is bounded by $W_2(\tilde{\eta}_t, \hat{\eta}_t)^2$. The marginal of $|\tilde{\boldsymbol{x}} - \hat{\boldsymbol{x}}|^2$ is bounded by the same quantity, since the $\mathcal{X}$-projection of $\gamma^*$ couples the $\boldsymbol{x}$-marginals of $\tilde{\eta}_t$ and $\hat{\eta}_t$. We arrive at

$$W_2(F_\sharp \tilde{\eta}_t, F_\sharp \hat{\eta}_t)^2 \leq \int |F(\tilde{\boldsymbol{x}}, \tilde{\boldsymbol{\xi}}) - F(\hat{\boldsymbol{x}}, \hat{\boldsymbol{\xi}})|^2\, \mathrm{d}\gamma^*\big((\hat{\boldsymbol{x}}, \hat{\boldsymbol{\xi}}), (\tilde{\boldsymbol{x}}, \tilde{\boldsymbol{\xi}})\big) \leq 2\,(1 + \mathrm{d}t^2 L_R^2)\, W_2(\tilde{\eta}_t, \hat{\eta}_t)^2. \tag{32}$$

Substituting this into (23) in place of the Lipschitz bound and proceeding as before yields

$$\mathbb{E}\, W_2(\hat{\rho}_{t+1}, \tilde{\rho}_{t+1})^2 \leq C\,(1 + \mathrm{d}t^2 L_R^2)\,\min(M, \tilde{M})^{-2/\alpha}. \tag{33}$$

*SDE form.* For $F(\boldsymbol{x}, \boldsymbol{\xi}) = \boldsymbol{x} + \mathrm{d}t\, b(\boldsymbol{x}) + \varepsilon\, \sigma(\boldsymbol{\xi})$ with $b$ and $\sigma$ Lipschitz with constants $L_b, L_\sigma$, we use the three-term inequality $|a + \mathrm{d}t\, b + \varepsilon\, c|^2 \leq 3\,(|a|^2 + \mathrm{d}t^2 |b|^2 + \varepsilon^2 |c|^2)$ and bound, under $\gamma^*$,

$$|F(\tilde{\boldsymbol{x}}, \tilde{\boldsymbol{\xi}}) - F(\hat{\boldsymbol{x}}, \hat{\boldsymbol{\xi}})|^2 \leq 3\,|\tilde{\boldsymbol{x}} - \hat{\boldsymbol{x}}|^2 + 3\mathrm{d}t^2 L_b^2 |\tilde{\boldsymbol{x}} - \hat{\boldsymbol{x}}|^2 + 3\varepsilon^2 L_\sigma^2 |\tilde{\boldsymbol{\xi}} - \hat{\boldsymbol{\xi}}|^2. \tag{34}$$

Proceeding as in the corollary,

$$\mathbb{E}\, W_2(\hat{\rho}_{t+1}, \tilde{\rho}_{t+1})^2 \leq C\,(1 + \mathrm{d}t^2 L_b^2 + \varepsilon^2 L_\sigma^2)\,\min(M, \tilde{M})^{-2/\alpha}. \tag{35}$$

$\square$

## B. Explicit construction of an interpolating map

Assume $\boldsymbol{\xi}_t$ is distributed uniformly on the unit sphere $\mathbb{S}^{d-1}$ and $d \geq M$. Consider the following general form for $F$:

$$F(\boldsymbol{x}_t, \boldsymbol{\xi}_t) = b(\boldsymbol{x}_t) + \sum_{i,k=1}^{M} (\hat{\boldsymbol{x}}_{t+1}^i - b(\boldsymbol{x}_t))\hat{\mathbb{G}}_{ik}^{-1}\hat{\boldsymbol{\xi}}_t^k \cdot \boldsymbol{\xi}_t, \tag{36}$$

where $b : \mathcal{X} \to \mathcal{X}$ captures the mean behavior, i.e. $b(\boldsymbol{x}_t) = \mathbb{E}\left[\boldsymbol{X}_{t+1}|\boldsymbol{X}_t = \boldsymbol{x}_t\right]$, $(\hat{\boldsymbol{x}}_{t+1}^i, \hat{\boldsymbol{\xi}}_t^i)$, $i = 1, \ldots, M$ are the training points, and $\hat{\mathbb{G}}_{ik} = \hat{\boldsymbol{\xi}}_t^i \cdot \hat{\boldsymbol{\xi}}_t^k$. Note that $\hat{\mathbb{G}}$ is almost surely invertible when $d \geq M$.

This function interpolates the training data:

$$F(\hat{\boldsymbol{x}}_t^j, \hat{\boldsymbol{\xi}}_t^j) = b(\boldsymbol{x}_t) + \sum_{i,k=1}^{M} (\hat{\boldsymbol{x}}_{t+1}^i - b(\boldsymbol{x}_t))\hat{\mathbb{G}}_{ik}^{-1}\hat{\boldsymbol{\xi}}_t^k \cdot \boldsymbol{\xi}_t^j \tag{37}$$

$$= b(\boldsymbol{x}_t) + \sum_{i=1}^{M}(\hat{\boldsymbol{x}}_{t+1}^i - b(\boldsymbol{x}_t))\delta_{ij} = \hat{\boldsymbol{x}}_{t+1}^j \quad \forall j = 1, \ldots, M \tag{38}$$

At the same time, it is an affine function in $\boldsymbol{\xi}_t$ and as regular in $\boldsymbol{x}_t$ as $b$ is, and the latter is smooth in physical systems and natural videos. We can improve the regularity of $F$ by improving the conditioning of $\hat{\mathbb{G}}$. As discussed, $\hat{\mathbb{G}}$ approaches the identity matrix for large $d$ and fixed $M$.

Furthermore, note that for any $\mathbf{v} \in \mathbb{S}^{d-1}$, $\sqrt{d}\mathbf{v} \cdot \boldsymbol{\xi}_t \to \eta \sim \mathcal{N}(0,1)$ in distribution as $d \to \infty$ (Vershynin, 2018)[Theorem 3.3.9]. For fixed $\boldsymbol{x}_t$, the distribution of $F(\boldsymbol{x}_t, \boldsymbol{\xi}_t)$ is thus similar to

$$b(\boldsymbol{x}_t) + \frac{1}{\sqrt{d}}\sum_{i=1}^{M}(\hat{\boldsymbol{x}}_{t+1}^i - b(\boldsymbol{x}_t))\eta_i \quad \text{where} \quad \eta_i \sim \mathcal{N}(0,1)\,\forall i\,, \tag{39}$$

namely a Gaussian perturbation of the expected value of $\boldsymbol{X}_{t+1}|\boldsymbol{X}_t = \boldsymbol{x}_t$ in the directions of other training points. We note that this explicit affine construction is illustrative only and is not the model optimized in our experiments. A general $F_\theta$ does not need to be affine in $\boldsymbol{\xi}$ and can transform Gaussian into non-Gaussian conditional outputs as our experiments show. Thus, (39) should *not* be interpreted as a restriction of Stochastic Lifting to Gaussian conditional marginals.

## C. Nearest Neighbor analysis

We conduct a nearest neighbor analysis to ensure that our model generates novel samples rather than memorizing training data. Specifically, for each example: the top row shows a randomly selected generated sample, the middle row displays the closest matching sample from the training set, and the bottom row shows the closest training-set neighbor to the middle row itself. The visual variation between the generated sample (top row) and its nearest training-set neighbor (middle row) is similar in scale to the variation observed between the two training-set samples (middle and bottom rows). This demonstrates that our model generalizes effectively without simply replicating training examples.

## D. Numerical experiments

### D.1. Duffing SDE

The physical trajectories $x_t \in \mathbb{R}^2$ follow the SDE with

$$\frac{\mathrm{d}}{\mathrm{d}t}\begin{bmatrix} x_1 \\ x_2 \end{bmatrix}(t) = \begin{bmatrix} x_2 \\ -0.4x_2 + x_1 - 0.2(x_1)^3 \end{bmatrix}(t) + \begin{bmatrix} 0 \\ 1 \end{bmatrix}\xi(t). \tag{40}$$

The initial configuration is a Gaussian centered at $m_0 = [0, 10]$ with diagonal variance of strength 0.5. The dynamics are integrated over $0 \leq t \leq 14$ for $N = 5000$ samples.

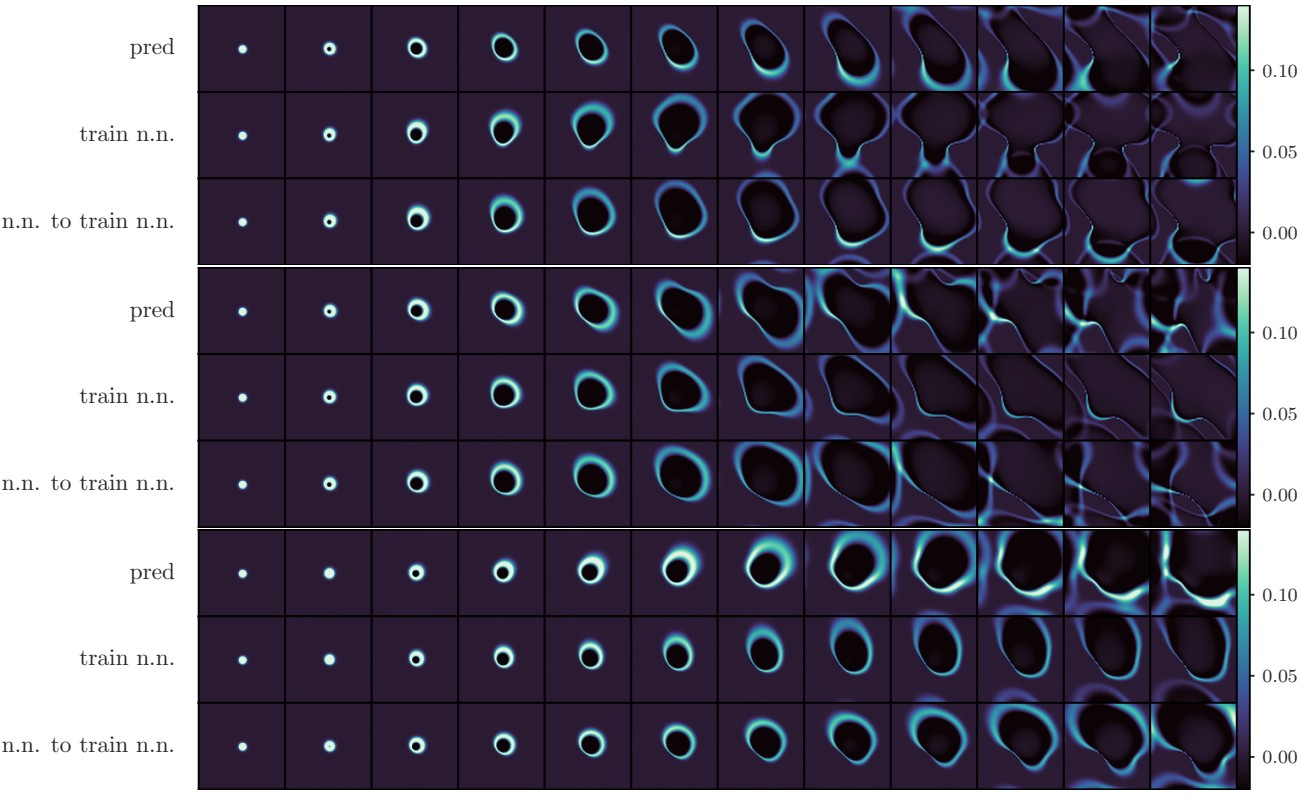

*Figure 7.* **Wave nearest neighbor analysis**. Of each image: top row randomly chosen generated sample, middle row nearest neighbor in the training set, bottom row nearest neighbor in the training set to the middle row. The variation between the top and middle row should be roughly similar to the variation between the middle and bottom row, indicating generalization without memorization.

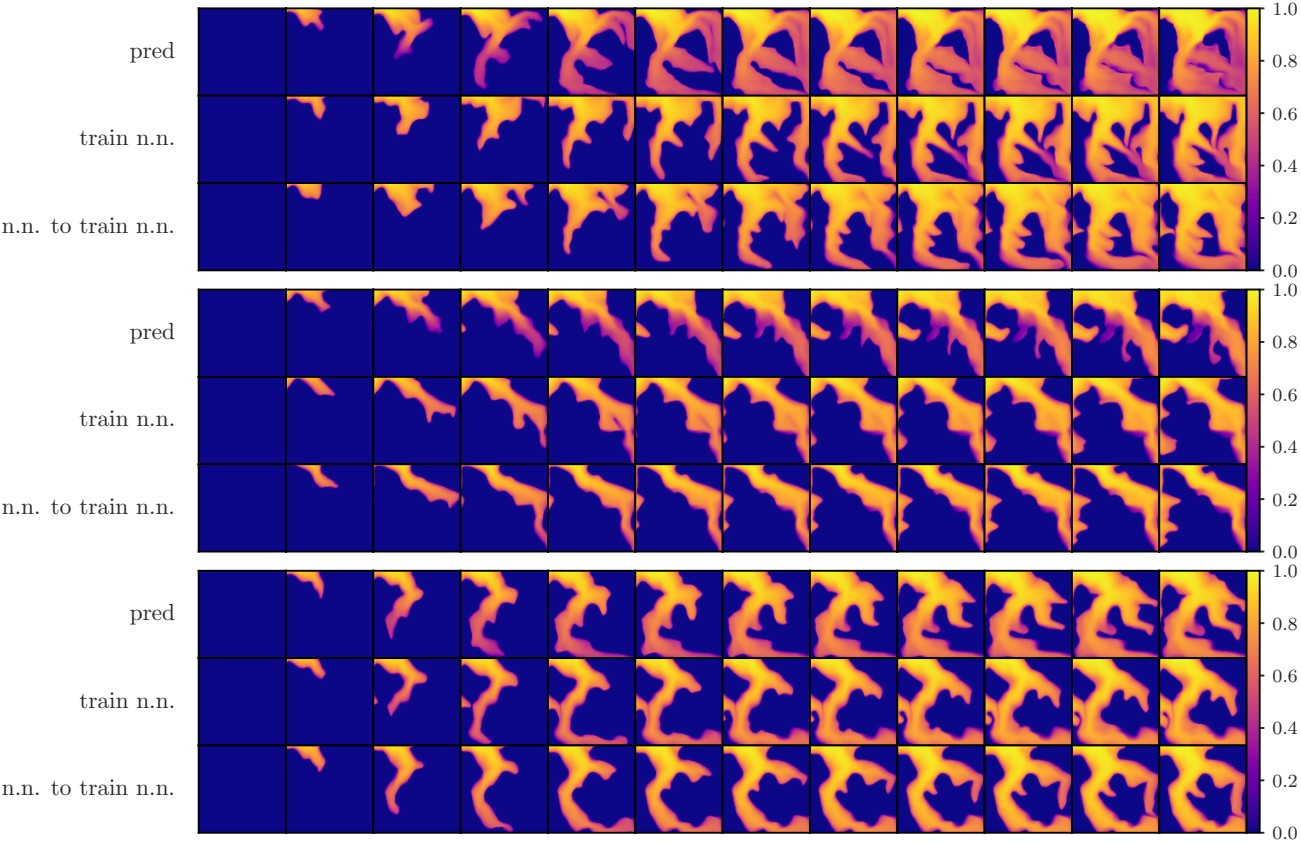

*Figure 8.* **Flow nearest neighbor analysis**. Of each image: top row randomly chosen generated sample, middle row nearest neighbor in the training set, bottom row nearest neighbor in the training set to the middle row. The variation between the top and middle row should be roughly similar to the variation between the middle and bottom row, indicating generalization without memorization.

## D.2. Traveling wave through random media (Wave)

The initial condition is a Gaussian bump at the center of the domain. As the wave evolves, the random medium leads to different wave speeds that lead to a deformation of the wave front, which leads to a distribution of diverse solution fields; see Appendix G.1 for visualizations. Importantly, for both wave and flow problem, the initial condition is deterministic and fixed, thus deterministic approaches such as neural operators cannot capture the stochasticity induced by the random media and permeability fields during the autoregressive roll out; see Section 2.2.

Problem setup:

$$
\begin{cases}
\partial_{tt} u(t, \boldsymbol{x}) = c^2(\boldsymbol{x}) \Delta u(t, \boldsymbol{x}), & t \in (0, 8], \ \boldsymbol{x} \in \Omega, \\
u(0, \boldsymbol{x}) = \exp\left(-30 \left\| \boldsymbol{x} - [\pi, \pi] \right\|^2\right), & \boldsymbol{x} \in \Omega, \\
\partial_t u(0, \boldsymbol{x}) = 0, & \boldsymbol{x} \in \Omega.
\end{cases}
$$

$\Omega = [0, 2\pi] \times [0, 2\pi], T = 8.0$ We look at the 2D wave equation with periodic boundary conditions.

The wave speed $c(\boldsymbol{x})$ varies in space; it is generated by choosing random Fourier modes from a log-normal distribution which peaks at some wave number $k = 1$ which controls the regularity of the random field.

We have $M = 1024$ trajectories, discretized at $T = 64$ time points and $64 \times 64$ points in space.

## D.3. Two-phase flow in random porous media (Flow)

We consider an incompressible and immiscible two-phase flow system of water and oil phases (Aarnes et al., 2007). One phase of the flow moves through the domain as it interacts with a random permeability field which deforms the flow into a high variance distribution of complex shapes; see appendix G.2. The governing equations are

$$
-\nabla \cdot (K \lambda(s) \nabla p) = q \,,
$$

$$
\phi \frac{\partial}{\partial t} s + \nabla \cdot (f(s) v) = \frac{q_\omega}{\rho_\omega} \,,
$$

where $s$ is the saturation field, $K$ represents the permeability tensor, $\lambda(s)$ is the total fluid mobility (the sum of water and oil mobilities), and $q$ is the source term. In the second equation, $\phi$ denotes the porosity, $f(s)$ the fractional flow of water, and $v$ is the Darcy velocity.

The randomness enters via the permeability field $K$ which is given by the exponential $K(x) = \exp(\epsilon Z(x) + \bar{Z} + Y(x))$, where $Z$ and $Y$ are independent Gaussian random fields, each generated with a Matérn covariance function (marginal standard deviation $\sigma = 3.0$, correlation length $\lambda = 10$, smoothness parameter $\nu = 1$, noisiness parameter $\epsilon = 0.01$).

The equations are discretized with an explicit upwind finite-volume discretization method. We use $100 \times 100$ cells, end time 0.7, and time-step size 0.007. The initial condition is zero saturation with a localized source on the upper left corner of the spatial domain.

## D.4. Video data sets

We measure inference runtime by using the largest batch size that fits into the memory of our H100 GPUs and then divide the total runtime by that batch to get the per-video generation runtime.

## D.5. Metrics

For our physics-based examples, we regard the realizations of our stochastic process as time-dependent solution fields. That is, the $j$-th component of a sample $\boldsymbol{x}_t$ corresponds to the evaluation of $u : \mathcal{T} \times [0, L]^2 \to \mathbb{R}$ at the $j$-th grid point in $[0, L]^2$ and at time $t$.

To ease comparisons, we normalize $\mathcal{T}$ to $[0, 1]$ and $L$ to 1 when computing all metrics.

For the sake of comparison and visualization, we require low-dimensional metrics. We construct them as follows: Take a map $f : \boldsymbol{x}_t \mapsto f(\boldsymbol{x}_t) \in \mathbb{R}$. The push-forward of the empirical distribution of $\boldsymbol{x}_t$ under $f$ defines a distribution on $\mathbb{R}$.

Given samples from the ground truth $\boldsymbol{x}_t \sim \hat{\rho}_t$ and generated samples $\tilde{\boldsymbol{x}}_t \sim \tilde{\rho}_t$, the mismatch can be described by $W_2(f_\sharp \hat{\rho}_t, f_\sharp \tilde{\rho}_t)$, which is easy to compute as these are one-dimensional distributions.

The mismatch in this case is defined as a function of time $t$. It is also possible to consider a map $g$ that takes $\{\boldsymbol{x}_t\}_t$ an entire trajectory and returns a one-dimensional quantity of interest.

**Wasserstein of integrated mass**   We look at Wasserstein of integrated mass (WIM). In this case, the function $f$ corresponds to an $l_1$ norm, denoted $m$:

$$m(\boldsymbol{x}_t) = \frac{1}{n} \|\boldsymbol{x}_t\|_1 \approx \int_{[0,L]^2} |u(t,x)| \, \mathrm{d}x, \quad \text{WIM} = \frac{1}{T} \sum_{t=1}^{T} W_2(m_\sharp \hat{\rho}_t, m_\sharp \tilde{\rho}_t).$$

**Wasserstein crossing time**   Given some subset $B \subset [0,L]^2$ the earliest arrival time $\tau$ for some threshold value $c \in \mathbb{R}$ is given by

$$\tau(u) = \min\{t \in \mathcal{T} : u(t,x) > c \text{ for some } x \in B\}.$$

For a trajectory $\{\boldsymbol{x}_t\}_t$, this corresponds to a threshold value being crossed for a subset of entries of $\boldsymbol{x}_t$. Denote by $\varrho$ the path measure corresponding to trajectories $\{\boldsymbol{x}_t\}_t$. The Wasserstein crossing time is defined as

$$\text{WCT} = W_2(\tau_\sharp \hat{\varrho}, \tau_\sharp \tilde{\varrho}).$$

## E. Architecture details

**UNet backbone**   Our network follows the canonical encoder–decoder UNet with skip–connections, implemented in `Flax`. All architecture decisions are standard. We use GroupNorm, no dropout, stride-2 3×3 convolution. At each spatial scale for our residual blocks the channel depths are,

$$\texttt{medium\_feature\_depths} = [128, 256, 512],$$
$$\texttt{large\_feature\_depths} = [128, 256, 512, 1024].$$

The only architectural modification is the conditioning on the label $\boldsymbol{\xi}_t$. Many diffusion backbones condition on the "diffusion time" via some modulation scheme: The only difference is we skip the initial sinusoidal embedding which maps the diffusion-time scalar to a vector. Instead we directly embed $\boldsymbol{\xi}$ with a two-layer MLP with width $512$. The MLP output then modulates the residual feature maps via a standard FiLM modulation scheme (Perez et al., 2018).

As stated in the main text, apart from re-purposing the timestep embedding to encode the conditioning label $\xi$, all components adhere to established UNet practice, enabling a like-for-like comparison with standard diffusion backbones.

## F. Training details

For the physics-based problems (wave, flow), we use the L2 loss defined in equation 8. For the video datasets, we replace the L2 metric ($\|x - y\|_2^2$) with a perceptual-based LPIPS loss (Zhang et al., 2018) which we find improves FVD scores. This normalization prevents the network's predictions from diverging during rollout, making it particularly suitable for video data, which naturally lies within $[0,1]$.

*Table 3.* Hyper-parameters used for each dataset.

| | Wave | Flow | BAIR | CLEVRER |
|---|---|---|---|---|
| **Batch Size** | 64 | 64 | 128 | 32 |
| **Label Dim** | 64 | 64 | 512 | 256 |
| **Iterations** | 350 000 | 350 000 | 500 000 | 500 000 |
| **Optimizer** | AdamW | AdamW | AdamW | AdamW |
| **Learning Rate** | $1e^{-4}$ | $1e^{-4}$ | $1e^{-4}$ | $1e^{-4}$ |
| **Schedule** | cosine | cosine | cosine | cosine |
| **Network Size** | Medium | Medium | Medium | Medium |
| **Loss** | L2 | L2 | LPIPS | LPIPS |
| **Trajectories** ($M$) | 1024 | 1000 | 43 000 | 10 000 |
| **Time Points** ($T$) | 64 | 50 | 16 | 16 |
| **State Dim** ($n$) | $64^2$ | $96^2$ | $64^2 \times 3$ | $128^2 \times 3$ |
| **Markov Window** ($m$) | 3 | 3 | 1 | 3 |

# G. Uncurated Samples

## G.1. Wave

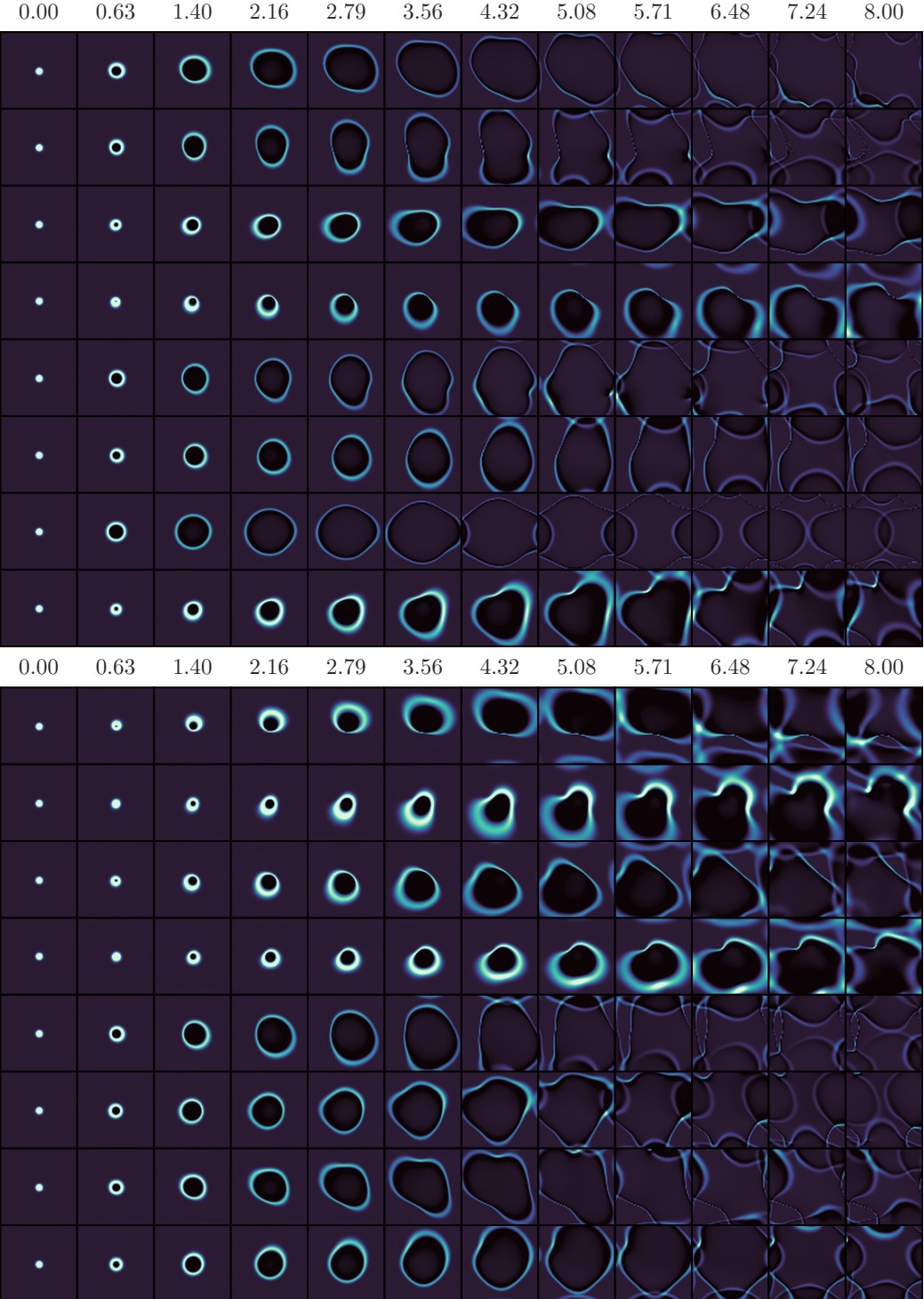

Figure 9. **Wave samples (uncurated)**. **Top**: true samples. **Bottom**: generated from Stochastic Lifting.

### G.2. Flow

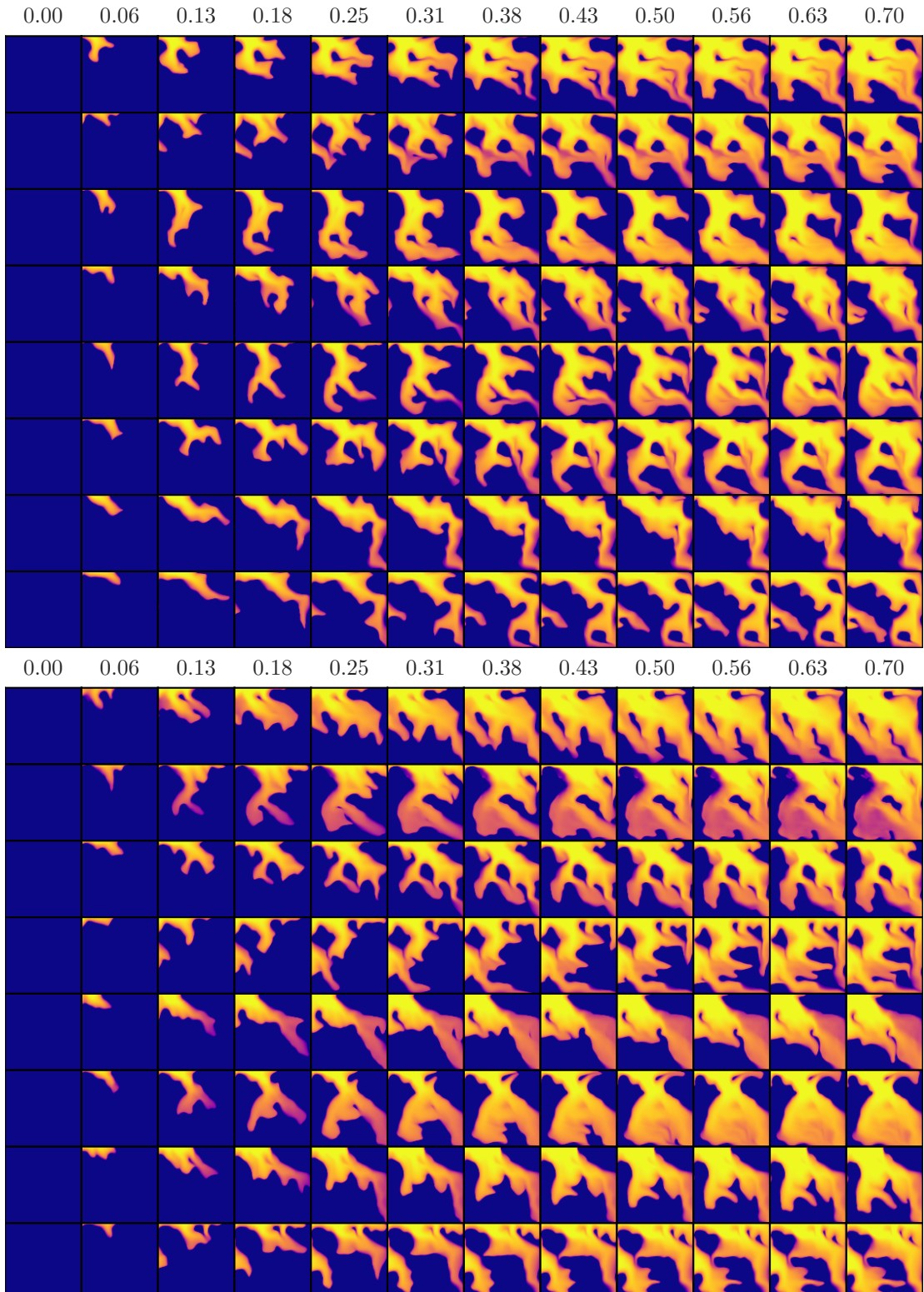

*Figure 10.* **Flow samples (uncurated)**. **Top**: true samples. **Bottom**: generated from Stochastic Lifting.

### G.3. CLEVRER

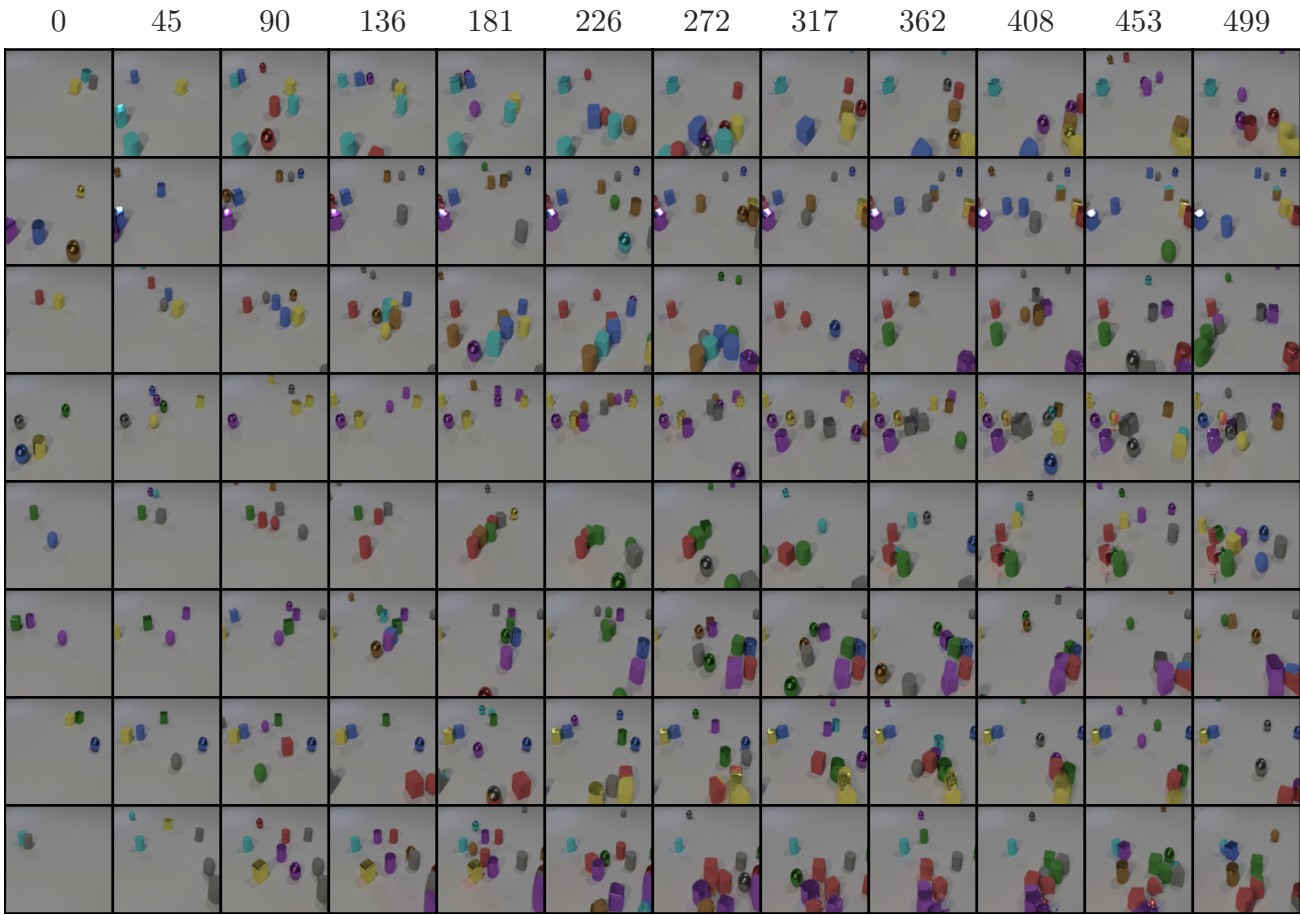

*Figure 11.* **CLEVRER samples (uncurated)**. Generated from Stochastic Lifting. Long rollout.

