# OpenReview forum: "Stochastic Lifting for Generating Trajectories of Stochastic Physical Systems"
_ICML.cc/2026/Conference — ICML 2026 regular_

### Official Review · Reviewer_y3je · 2026-03-12

**Soundness:** 2
**Presentation:** 2
**Significance:** 3
**Originality:** 2
**Overall Recommendation:** 3
**Confidence:** 3

**Summary:**

This paper has the goal to learn a map $F$ via regression that maps states of a stochastic process $x_t$ to future states $x_{t+1}$. Because of the process' stochasticity this constitutes learning a one to many mapping. The authors therefore propose "stochastic lifting", i.e. introducing an auxiliary random latent $\xi_t$ and learning the map $x_{t+1} = F(x_t, \xi_t)$ via regression. The method is supported by theoretical results: a) The expected Wasserstein distance between the training trajectories and the generated trajectories decays with the number of generated trajectories $M^{-1/\alpha}$. b) The constant of the bound depends in a favourable way on the time step size and dimension of the latent $\xi_t$ when the stochastic process is Lipschitz continuous in $t$. The method is tested on wave and fluid equations in random media, as well as video with random collision events.

**Compliance With Llm Reviewing Policy:**

Affirmed.

**Final Justification:**

While I don't think full agreement between theory and methodology is typically possible in machine learning, I think theory and methodology should meaningfully complement each other. In my opinion, this was not the case in the submitted version. However, I can envision substantial improvements from the authors rebuttal. I therefore believe that the paper would benefit from revision and resubmission, but I don't want to stand in the way to acceptance with my earlier score, and increase it to 3.

**Key Questions For Authors:**

1. What is the main message that the theoretical results should convey? How are they important to this work, and what is their significance to machine learning research?
2. Why do you pick the stochastic process to be bounded to the domain $[0, 1]^n$?
3. What is the definition of intrinsic dimension of data $\alpha$?

**Limitations:**

Yes

**Strengths And Weaknesses:**

I have a hard time speaking to the originality of the approach. Introducing a latent auxiliary variable to handle one to many relationships is a known technique in self-supervised learning. The application of this idea to stochastic processes is natural. However, I am mostly aware of works that attempt to learn the conditional or joint distributions between $(x_t, x_{t+1})$, instead of a deterministic map. Neural processes come to mind as a potentially related work not mentioned in the literature review. I would also expect that similar ideas have been thought about in the context of state space models for stochastic processes, possibly enforcing more structure on the latent $\xi_t$.

Strengths:
- The method is simple and easy to implement
- The method shows promising results on the presented benchmarks, and the experiments are interesting

Weaknesses:
- The paper mixes two incompatible notions. The theory focusses on smooth processes, and smoothness is mentioned multiple times in the contributions (contribution (2)) and in the discussion of assumptions for the theoretical results. At the same time, stochastic processes that depend on Brownian motion $W_t$ are stated explicitly as example setting. However, Brownian motion is known to be nowhere differentiable. In fact, the assumption that $x_{t+1} -x_t \in O(\Delta t)$ is violated for Brownian motion, for which, heuristically speaking, $W_t - W_s \in O(\sqrt{t-s})$ would be more accurate.
- The smoothness assumption is made to obtain stronger theoretical guarantees. However, the bound in Corollary 3.2. contains an irreducible constant error term "3C" which is not further explored. Therefore, Corollary 3.2. is only of interest whenever the time steps are of the same order as this constant. We could drop the smoothness assumption and keep most of the theoretical implications intact.
- The same critique applies to Proposition 3.3.. The dimension of the latent only controls one of the two contributions to the Lipschitz constant, and competes with the dimension of the stochastic process. There would be other ways to achieve a small Lipschitz constant. If we, for example, pick latent vectors from a sphere with diameter D, the Lipschitz constant would decay like diam(X)/D, a stronger decay than the one stated in the paper, but I doubt it'd be a useful one.
- Methodologically, the points above are not a major issue. However, the authors put great emphasis on the theoretical results. The emphasis on theory does not match the rigour applied or the insight gained.
- The significance of the theoretical results is not obvious to me from the writing of the paper.

---

> ### Author Rebuttal · Authors · 2026-03-30
>
> We thank the reviewer for taking the time to go through the details of our work. We appreciate pointing out the areas of uncertainty and address each point below.
>
> —
>
> ***"The paper mixes two incompatible notions. The theory focuses on smooth processes … At the same time, stochastic processes that depend on Brownian motion W are stated explicitly as example setting. However, Brownian motion is known to be nowhere differentiable [...]"***
>
> - We appreciate this point but believe there is a misunderstanding. Our smoothness assumptions apply to the *discrete-time transition map*, not to the continuous Brownian path.  The nowhere-differentiability of Brownian sample paths is a property of the *continuous-time limit* and does not affect the regularity of the finite-step transition kernel, which is the object we learn.
> - The framework of stochastic lifting assumes a strong conditioning on the previous frame is possible – in other words, it assumes noise that is small enough relative to the time-step size, i.e. scale of noise $\leq \sqrt{\mathrm dt}$. We agree with the reviewer that SL is not a good method to learn pure Brownian motion.
> - The scaling ($\mathcal O(\sqrt{\mathrm dt})$) is covered in Appendix B.2 (in particular, equation (19) with $\varepsilon = \sqrt{\mathrm dt}$. We will make this more clear in the revision.
>
> ***"The same critique applies to Prop. 3.3.. [...] I doubt it'd be a useful one."***
>
> - We agree that inflating the label norm would trivially reduce the Lipschitz constant on paper. However, this does not improve the bound in Proposition 3.1: rescaling labels by $D$ reduces $L_F$ by $1/D$ but the map must now operate over an input domain of diameter $D$, and the generalization behavior of neural networks degrades with input scale. We assume throughout the presentation that data is normalzed as stated in Proposition 3.3. The advantage of high-dimensional ***unit-norm*** labels (Proposition 3.3) is that separation is achieved through near-orthogonality without increasing input magnitude, keeping the learning problem well-conditioned. We will add a remark contrasting these two mechanisms.
>
> ***"The authors put great emphasis on the theoretical results. The emphasis on theory does not match the rigour applied or the insight gained."***
>
> - We appreciate this feedback and agree that the current framing may overstate the role of theory. We will rebalance the presentation in the revision.
> - To clarify what we believe the theory does and does not do: it is not intended as a complete explanation of why Stochastic Lifting works. It serves three specific roles. First, it identifies smoothness as the key property that makes one-step sampling from an interpolating map accurate (Proposition 3.1), which is not obvious since interpolation alone could produce arbitrarily poor samples. Second, it provides actionable guidance: increasing the label dimension improves separability and reduces the minimal Lipschitz constant (Proposition 3.3), telling practitioners how to set d. Third, it explains why the transition-learning setting is favorable: the small-perturbation structure of consecutive states further reduces the effective Lipschitz constant (Corollary 3.2), clarifying why the method works for trajectory data but not for noise-to-image generation.
> - That said, we consider a large portion of our contribution to be empirical. We demonstrate state-of-the-art one-step trajectory generation across qualitatively different domains (stochastic PDEs and video) and show that the method reproduces distributional quantities of interest.
> - We will revise the paper to foreground the empirical contributions and reframe the theory as qualitative insight and practical guidance.
>
> ***"Introducing a latent auxiliary variable to handle one to many relationships is a known technique … Neural processes come to mind as a potentially related work"***
>
> - We agree that auxiliary latent variables are broadly used. The distinction is that neural processes and related methods typically *infer* the latent from context data (encoder), whereas Stochastic Lifting draws labels independently and relies on the high-dimensional geometry for separation - no encoder or posterior inference is needed. We will add a discussion of neural processes and state-space models to the related work, clarifying this architectural and conceptual difference.
>
> ***"Why do you pick the stochastic process to be bounded to the domain $X = [0,1]^n?$"***
>
> - This is a standard normalization assumption for the theoretical analysis that simplifies the constants in the Wasserstein bounds. It can be relaxed to any compact domain by rescaling; the key quantities (Lipschitz constant, intrinsic dimension) are scale-invariant up to constants.
>
> ***"What is the definition of the intrinsic dimension of data?"***
>
> - We use the Wasserstein intrinsic dimension as defined in [Weed & Bach, 2019], in particular Definition 4 therein. We will add this definition explicitly in the main text.

---

> > ### Author Rebuttal · Reviewer_y3je · 2026-04-03
> >
> > I thank the authors for their thorough explanations. Many of my remarks were responded to in the rebuttal, but I am not fully convinced that the authors explained well how theory and methodology are connected in this paper. Specifically, the paper states prominently (173, right column)
> > > even stochastic physics systems lead to trajectories that gradually change
> > over time in the sense that ∥xit+1−xit∥= O(dt) scales with a small time-step size dt for all times and realizations.
> >
> > This statement from the paper does not express that the limit $dt\to 0$ was excluded from the theory, and that only finite time transitions are being considered in the paper. Meanwhile the rebuttal states
> > > Our smoothness assumptions apply to the discrete-time transition map, not to the continuous Brownian path.
> >
> > I think this is an example of the conceptual **clarity** that I was missing in the original paper. The first statement is violated for large classes of stochastic processes, and it also conveys a totally different regularity assumption than the point made by the authors in the rebuttal. I believe I have a better understanding of what the authors hoped to convey in the original paper. To help with my understanding:
> > - Could the authors clarify how they would define smoothness for discrete time transition maps?
> > - Are processes like $F(x_t, \xi) = x_t + sqrt(dt)\xi$ captured by the theory or not? (Note that the sqrt function is not Lipschitz at zero, and L_F blows up for small time steps)

---

> > > ### Author Response · Authors · 2026-04-03
> > >
> > > Thank you for this helpful follow-up.
> > >
> > > ***Could the authors clarify how they would define smoothness for discrete time transition maps?***
> > >
> > > By “smoothness” in the paper we mean Lipschitz regularity of the discrete-time transition map at a fixed finite time step *on the lifted data*. That is, for a given $dt$, we assume the next state can be represented as
> > >
> > > $$ X_{t+1} = F(X_t,\xi_t), $$
> > >
> > > and $F$ is regular in the sense that
> > >
> > > $$ \|F(x,\xi) - F(x',\xi')\| \leq L_{F}\|(x,\xi) - (x',\xi')\|. $$
> > >
> > > This is the notion stated and used in Proposition 3.1. So the relevant object is the finite-step map $F$, not the regularity of continuous Brownian trajectories.
> > >
> > > We agree with the reviewer that the sentence about
> > >
> > > $$\ \|x_{t+1}^i - x_t^i\| = O(dt)$$ for all times and realizations was stated too broadly, and in this form it is misleading exactly because as the reviewer points out it conflates two types of smoothness. As the reviewer correctly states, if $x_t$ here are the non-lifted states of (e.g.) Brownian motion, the result would be $O(\sqrt dt)$. We will remove this statement and replace it with the precise finite-step map smoothness that we build on.
> > >
> > > ***This statement from the paper does not express that the limit $dt \to 0$ was excluded from the theory, and that only finite time transitions are being considered in the paper.***
> > >
> > > This is correct. We will make this completely explicit in the revision.
> > >
> > > Our problem setting is discrete-time throughout: we assume access to data already discretized in time, and we learn a one-step map between successive observed states. We do **not** analyze the continuous-time limit $dt \to 0$, and we do not make claims about pathwise regularity of the underlying continuous-time process.
> > >
> > > ***Are processes of the form $X_{t+1} = X_t + \sqrt{dt} W_t$ captured by the theory or not?***
> > >
> > > In principle, yes our theory accommodates this at the level of **fixed-dt discrete-time transitions** (line 191, right column). The key point is that we do not learn the continuous-time Brownian path. We learn the discrete-time transition map at a fixed step size $dt$.
> > >
> > > In practice, we do not aim to learn systems with no drift as we rely on the strong state coupling given by the drift term. No drift would be analogous to the case where we take $dt \to 0$. See discussion below.
> > >
> > > ***On the limit of $dt \to 0$***
> > >
> > > The reviewer points out an important point in the case that the noise scales with $\sqrt{dt}$ while the drift scales with $dt$. In the limit the noise will dominate and there will be no strong coupling between state pairs $x_t$ and $x_{t+1}$
> > >
> > > But we note this is not a fatal issue because, we often operate in a regime where **on the time-discrete data** the scale of the noise is meaningfully smaller than the scale of the drift. Our empirical success reflects this.
> > >
> > > The fact that the scale of the noise is limited shows in Equation (35) in the Appendix, specifically the term $\varepsilon / dt$ for the case $F(x, \xi) = x + b(x) dt + \varepsilon \sigma( \xi)$.
> > >
> > > We explicitly show we rely on this coupling in Figure 2 where we examine the effect of shuffling the trajectories: as we state in the caption "shuffling training pairs $(x_t, x_{t+1})$ breaks the conditional transition structure and degrades Stochastic Lifting’s performance". In terms of our theory, this increases the quantity $L(D_\xi)$ given in Eq. (7) by increasing the numerator of the fraction.
> > >
> > > We will add this additional discussion to our revision.
> > >
> > > Thank you again for pressing on this point. We believe this clarification substantially improves the connection between the theory and the methodology, and we hope these changes address your remaining concern.

---

### Official Review · Reviewer_op4e · 2026-03-12

**Soundness:** 2
**Presentation:** 3
**Significance:** 3
**Originality:** 3
**Overall Recommendation:** 4
**Confidence:** 4

**Summary:**

The paper introduces "Stochastic Lifting" to generate trajectories of stochastic physical systems using a single neural network evaluation per time step. To avoid collapsing to the conditional mean, the authors append a high-dimensional, independently drawn random label $\xi_t$ to each training pair $(x_t, x_{t+1})$. This transforms a one-to-many stochastic transition into a deterministic one-to-one mapping. At inference, sampling a fresh random label and passing it through the trained network generates new trajectory states. Empirical results demonstrate the effectiveness of the proposed method.

**Compliance With Llm Reviewing Policy:**

Affirmed.

**Final Justification:**

Thanks for the authors' detailed response. My concerns have been adequately addressed. I am willing to adjust my score to 4.

**Key Questions For Authors:**

I am genuinely open to raising my score if the authors can address these fundamental gaps between the proposed theory and the empirical success.


- Given the collapse to the mean as $M \to \infty$ (Section 3.4), is it more accurate to describe Stochastic Lifting as a finite-sample regularization/interpolation technique rather than a true distribution-learning generative model? How do you mathematically justify expanding the purely artificial label dimension $d$ as the physical dataset $M$ grows?


- Auto-regressive rollout heavily compounds errors if the transition $\rho(\cdot | x_t)$ is wrong. Can you provide any theoretical intuition or justification for why the *conditional* dynamics are preserved, given that Proposition 3.1 only bounds the *marginal* distribution?


- Since your explicit mathematical construction (Appendix C) effectively yields a Gaussian perturbation, wouldn't a purely theoretical implementation of Stochastic Lifting fail on multi-modal physical transitions? To what extent is your empirical success actually relying on the implicit bias/regularization of U-Net and AdamW, rather than the theoretical properties of the high-dimensional lifting itself?

**Limitations:**

The authors discuss some practical limitations (e.g., failing on noise-to-image tasks). However, the fundamental theoretical limitations regarding statistical inconsistency (the infinite data collapse) and the disparity between the marginal theoretical bound and the conditional generative task are not discussed as limitations.

**Strengths And Weaknesses:**

**Strengths:**

- **Empirical Efficiency:** The one-step inference is practically appealing and computationally efficient. The empirical results on complex datasets (e.g., wave propagation, two-phase flow, and video benchmarks) appear highly competitive, especially given the low computational cost compared to multi-step diffusion models.


- **Simplicity:** The core mechanism is straightforward to implement and integrates well with existing architectures without requiring complex latent space encodings.

**Weaknesses:**

I appreciate the strong empirical results. However, my main concern is that the theoretical framework presented does not adequately explain *why* the method works in practice. In fact, the theory suggests fundamental limitations that make me suspect the empirical success is largely driven by the implicit regularization (inductive bias) of the U-Net architecture and optimization process, rather than the theoretical validity of Stochastic Lifting as a generative model.

- 1. A fundamental property of a generative model is that it should learn the true distribution better as data increases. However, the authors explicitly state in Section 3.4 that taking the infinite data limit $M \to \infty$ causes the model to collapse to the conditional expectation $\mathbb{E}[X_{t+1} | X_t = x_t]$. Scaling the noise dimension $d$ with the dataset size $M$ to prevent this collapse implies the method lacks a well-posed population limit.

- 2. The core task of trajectory generation is capturing the correct *conditional* dynamics $\rho(\cdot | x_t)$. Yet, the primary theoretical guarantee, as stated in Proposition 3.1, only bounds the Wasserstein distance between the *marginal* empirical measures. Matching the overall distribution of states at step $t+1$ provides no mathematical guarantee that the specific, auto-regressive transitions from $x_t$ are physically correct.


- 3. In Appendix C, the explicit mathematical construction of the interpolating map demonstrates that as $d \to \infty$, the output behaves as a Gaussian perturbation around the mean $b(x_t)$. If the theoretical framework constrains the model to a Gaussian distribution, it would likely struggle to generate complex, varied outputs. The fact that it *does not* fail empirically strongly suggests that the theoretical explanation is incomplete, and the U-Net's engineering is doing the heavy lifting.

---

> ### Author Rebuttal · Authors · 2026-03-30
>
> Thank you for the careful reading of the paper. We share a lot of the analysis regarding the relationship between our theory and empirical results.
>
> —
>
> ***"Is it more accurate to describe Stochastic Lifting as a finite-sample regularization/interpolation technique rather than a true distribution-learning generative model? How do you mathematically justify expanding the purely artificial label dimension d as the physical dataset M grows?"***
>
> - We agree with your characterization and believe it is a strength rather than a limitation: Stochastic Lifting is deliberately designed as a finite-sample interpolation method, not a distribution-learning generative model in the classical statistical sense. We will make this framing more explicit in the revision.
> - While there is an absence of a useful population limit; we note that for the regime we target (fixed, finite training trajectories from expensive simulators), the classical M→∞ limit is not operationally relevant. The scaling of d with M is analogous to how kernel methods scale feature dimension with sample size to control the bias-variance tradeoff.
> - We will add a discussion explicitly positioning SL within the finite-sample interpolation paradigm and contrasting it with distribution-learning approaches.
>
> ***"Proposition 3.1 only bounds the marginal distribution … Can you provide any theoretical intuition or justification for why the conditional dynamics are preserved?"***
>
> - This is a valid observation about the current theory. First, we acknowledge that a formal conditional bound remains open and we will state this explicitly as a direction for future theoretical work.
> - Heuristically, for the specific structure of transition learning, the marginal bound is evaluated on test pairs drawn jointly from the coupling between consecutive states, so it measures the quality of the joint empirical coupling at each step, not just the time-marginal in isolation.
>
> ***"I suspect the empirical success is largely driven by the implicit regularization (inductive bias) of the U-Net architecture and optimization process, rather than the theoretical validity of Stochastic Lifting."***
>
> - We agree that the inductive bias of modern architectures is essential, but this is true of all contemporary generative models, not a weakness specific to Stochastic Lifting. To confirm this, we trained both SL and ARDM with an MLP backbone: neither learned anything meaningful, as expected. What matters for Stochastic Lifting specifically is that the architecture's implicit bias enables learning a smooth interpolating map over the lifted data. The surprising and, we believe, exciting conclusion of our work is that in the regime we consider, a standard regression loss combined with a UNet is sufficient to learn this map, with no adversarial training, no multi-step sampling, and no distillation. Our theory provides intuition for why this works: high-dimensional labels reduce the minimal Lipschitz constant of interpolants and smooth interpolation yields accurate samples. The theory is certainly not the whole picture. A complete explanation would require a rigorous characterization of architectural implicit bias, which remains an open problem across deep learning. But the theory does identify the right target property (smoothness) and explains why lifting helps achieve it.
>
> ***The explicit construction (Appendix C) effectively yields a Gaussian perturbation.***
>
> - We include this example as an illustration what *could* be learned, but it is by no means claimed that this is what the optimization yields. Note that the explicit construction from Appx. C depends linearly on the label $\xi$. A more general map $F$ does not need to lead to Gaussian marginals, in the spirit of noise outsourcing, the Gaussian random input $\xi$ could be transformed to yield a different distribution.
>
> ***"The fundamental theoretical limitations regarding statistical inconsistency (the infinite data collapse) and the disparity between the marginal theoretical bound and the conditional generative task are not discussed as limitations."***
>
> - We will add both points explicitly to the limitations section: (i) Stochastic Lifting is a finite-sample method without a meaningful population limit, and (ii) the current theory bounds marginal rather than conditional quality. We believe stating these clearly strengthens the paper by setting appropriate expectations for the theoretical contribution.
> - Additionally we would like to clarify what we believe the theory does and does not do: it is not intended as a complete explanation of why Stochastic Lifting works. It serves three specific roles. First, it identifies the role of smoothness for accuracy. Second, it illustrates how large $d$ improve separability. Third, it explains why the transition-learning setting is favorable, clarifying why the method works for trajectory data but not for noise-to-image generation.

---

> > ### Author Rebuttal · Reviewer_op4e · 2026-04-02
> >
> > My concerns have been adequately addressed. The author has provided a detailed response and made necessary revisions. I will adjust my score to 4.

---

> > > ### Author Response · Authors · 2026-04-02
> > >
> > > We thank Reviewer KT84 for the constructive feedback that helped improve the paper, and for taking the time to carefully evaluate our response. We are glad the revisions and additional details addressed your concerns and you were able to raise your score.

---

### Official Review · Reviewer_q5by · 2026-03-12

**Soundness:** 4
**Presentation:** 2
**Significance:** 3
**Originality:** 2
**Overall Recommendation:** 5
**Confidence:** 3

**Summary:**

This paper considers the problem of learning the generation of stochastic trajectories. In real-world applications, these trajectories are often in unstructured, high-dimensional space (e.g. videos), requiring efficient, purely data-driven algorithms.

The authors propose a simple method: augmenting single observations with random features (fixed, only sampled once) and learning a neural map to overfit on predicting the immediate state transition. The accompanying theoretical analysis is supporting this approach, demonstrating, for example, its stability. These results are empirically verified on synthetic data.

The proposed method is compared to a range of one- and multi-step video-generation methods, relying on continuous-time generative modeling approaches. On real-world benchmark datasets, the proposed method vastly outperforms all one-step methods, and even many multi-step methods.

**Compliance With Llm Reviewing Policy:**

Affirmed.

**Final Justification:**

As stated in the acknowledgment, the rebuttal addressed my raised concerns well.

The authors view their work as an empirical justification of a bespoke, efficient solution to finite-sample problems with small pertubations in each step. In my view, the authors achieve their goal with the original experimental setup, and the additional discussion in their rebuttal.

Therefore, and according to my original review, I will recommend acceptance.

**Key Questions For Authors:**

1. What hyperparameters did you use for the baselines? In particular, did you train them with the same Markovian assumption? See also Weaknesses 1..
2. How do you decided when to stop training? Presumably, a standard monitoring of a validation set does not work, because of the random label augmentation.
3. What is your intuition or reasoning for why ARDM is so much worse than your method? Intuitively, with enough steps, ARDM should be as expressive as your method, yet there remains a performance gap.
4. Similarly for the multi-step methods. What is the intuition why your much simpler approach with Markovian modeling assumption can almost reach the performance of Rolling Diffusion, a optimized, expressive, multi-frame context diffusion approach?


I believe information of Questions 1 and 2 is essential to include in the manuscript. If strong arguments, or additional discussions / results, are provided for Questions 3 and 4 (and related Weaknesses 2 and 3), I will gladly consider recommending acceptance.

**Limitations:**

Yes.

**Strengths And Weaknesses:**

*Strengths:*
1. The proposed method is quite simple and elegant.
2. It performs exceptionally well on a range of synthetic and real-world datasets.
3. The theoretical analysis and its empirical validation is informative and insightful.


*Weaknesses:*
1. The experimental approach to the baselines in Tables 1 and 2 is missing, hindering reproducibility. In particular, hyperparameters, training details and their applications to the specific problems are not discussed.
2. The methodological gap to the baselines is large, in particular the Markovian assumption (1) to the (potential) multi-frame context video-generation models. Arguably the closest baseline, a autoregressive one- or few-step conditional continuous-time generative model with the same Markovian assumption, is not considered.
3. The extraordinary good performance (in Tables 1 and 2), compared even to most multi-step models, is not analyzed or justified. Given the methodological gap in 2., and the comparatively simple approach, it is difficult to interpret or understand where this performance boost stems from.

---

> ### Author Rebuttal · Authors · 2026-03-30
>
> We thank the reviewer for the thorough review. We will address the raised concerns below.
>
> —
>
> ***"The experimental approach to the baselines in Tables 1 and 2 is missing, hindering reproducibility. In particular, hyperparameters, training details and their applications to the specific problems are not discussed."***
>
> - We agree this information should be in the manuscript. We will add a dedicated "Baseline Details" paragraph.
> - For the ARDM baseline, we follow explicitly the set up of [Kohl et al., 2023] attempting to use all hyperparameters choices given in that paper as it is a standard reference for ARDMs applied to physics data. Importantly we specifically utilize the same UNet backbone and comparable parameter count as Stochastic Lifting. Otherwise, again following  [Kohl et al., 2023], the ARDM is trained with the same optimizer (Adam, β₁ = 0.9, β₂ = 0.999) and learning rate schedule (cosine lr = 1 × 10⁻⁴). The model conditions on 3 previous frames added along the channel dimension, uses a standard linear DDPM variance schedule.
> - We will include a table summarizing architecture, parameter count, optimizer, and training budget for every method.
>
> ***"Arguably the closest baseline, an autoregressive one- or few-step conditional continuous-time [...]  is not considered."***
>
> - We have now run exactly this comparison. On the wave and flow benchmarks, we trained two deterministic baselines with MSE loss: (i) a UNet Operator with the identical architecture, parameter count, and optimizer as Stochastic Lifting but with labels removed, and (ii) a standard Fourier Neural Operator (FNO). Both are rolled out autoregressively at test time.
> - The both models perform very poorly on the distributional metrics we consider because they cannot produce variation across multiple rollouts. Thus the W_2 is measured between the true distribution and a dirac distribution.
> - This confirms the claim in Section 2.2: without an explicit source of stochasticity, even expressive neural operators collapse to mean behavior and cannot recover distributional structure.
> - Additionally we evaluate both models on the BAIR video benchmark. We note in this case FVD is also a metric which takes into account distributional characteristics and the ability to generate diverse rollouts. The FNO models perform very poorly achieving an FVD of 142.0 and the UNet archives a FVD of 96.0. Both of these are significantly worse than our Stochastic Lifting which gets an FVD of 69.0
>
>
> ***"How do you decide when to stop training? [...]."***
>
> - We monitor distributional validation metrics (W₂ on quantities of interest for physics, FVD for video) evaluated with fresh labels, which remain well-defined despite the augmentation. Overfitting is not a significant issue in practice: the method aims for interpolation, and high-dimensional labels plus standard regularization (weight decay, normalization) ensure the learned map remains smooth at unseen labels. Early stopping helps but is not critical.
>
> ***"What is your intuition or reasoning for why ARDM is so much worse than your method? [...] ."***
>
> - Our intuition is twofold. First, ARDM must learn a score function that is accurate across all noise levels, and errors compound across diffusion steps  even with s=40, small per-step inaccuracies accumulate. Stochastic Lifting sidesteps this entirely: it learns a single direct map, so there is no sequential error accumulation within a time step.
> - Second, in the transition-learning setting, the target distribution $\rho(\cdot|x_t)$ is a small perturbation of the current state (Corollary 3.2), making it particularly well-suited to one-step generation. The diffusion process, by contrast, must traverse from pure noise to this narrow distribution regardless of how concentrated it is, which is an unnecessarily long path for a near-identity transition.
>
> ***"What is the intuition why your much simpler approach [...] Rolling Diffusion, an optimized, expressive, multi-frame context diffusion approach?"***
>
> - We clarify that Stochastic Lifting also conditions on multiple previous frames (as we state in the paper), so the comparison is not about context length. The key distinction is the generative mechanism. When consecutive states are close i.e., the transition law varies regularly and next states are small perturbations of the current state, a single-evaluation smooth map suffices for accurate samples. Our benchmarks operate in this regime, which is why a one-step approach nearly matches multi-step diffusion. When this premise is violated (e.g., large time steps), Stochastic Lifting degrades, as we show in Figure 2. Rolling Diffusion succeeds even with large time steps because multi-step sampling can represent more complex transitions outside our regularity regime. This is an intentional tradeoff: Stochastic Lifting trades generality for efficiency where the target distribution is close in a meaningful sense.

---

> > ### Author Rebuttal · Reviewer_q5by · 2026-04-02
> >
> > I have read the other reviews and rebuttals, and thank the authors for their clear replies.
> >
> > My concerns and questions have been addressed adequately.
> > I believe future readers will welcome the additional baselines and information about the experimental setup.
> > The authors may also consider adding some comments about the training procedure for clarity, and their interpretation of the performances of diffusion approaches to further distinguish their bespoke solution of the considered problem to these more general techniques.
> >
> > Throughout their rebuttals, the authors have affirmed their goal of providing an empirical evaluation of a relatively simple, efficient, targeted solution for finite-sample problems exhibiting only small pertubations in each step.
> > I believe their experiments demonstrate this effectively. However, given the slight confusion of some reviewers about this, the authors may consider reitterating these goals even more throughout the manuscript.
> >
> > The impact of this new method on the field should be judged by the community. Concerns about the theoretical foundations and results from the other reviewers may still be justified. Given that I did not bring up such concerns in my initial review, and the convincing replies of the authors to it, I will gladly recommend accepting this work, as promised.

---

> > > ### Author Response · Authors · 2026-04-02
> > >
> > > We thank Reviewer q5by for the thoughtful engagement throughout the review process. We will incorporate your suggestions on clarifying the training procedure and making the method's scope and goals more explicit throughout the manuscript. These revisions will improve the quality of the paper.
> > >
> > > We appreciate the recognition that the experiments speak to the method's effectiveness, and we share the view that its broader impact is something the community is well-positioned to evaluate. We thank the review for raising their score and recommending acceptance.

---

### Official Review · Reviewer_KT84 · 2026-03-13

**Soundness:** 3
**Presentation:** 2
**Significance:** 3
**Originality:** 3
**Overall Recommendation:** 4
**Confidence:** 3

**Summary:**

The authors introduce _Stochastic Lifting_, a novel paradigm for training models via regression losses for generating trajectories of stochastic physical systems in one step. At its core, the method uses random labeling of data points to produce an augmented dataset, where labels are drawn from a standard normal distribution. The augmented (randomly labeled) dataset is then used to train a one-step trajectory generator for physical systems. The authors provide theoretical support for their proposed method and conduct a thorough empirical evaluation across toy examples, physics datasets, and for the task of video generation.

**Compliance With Llm Reviewing Policy:**

Affirmed.

**Final Justification:**

The authors have provided a detailed response to my questions and concerns that I believe helps improve the quality of the paper. As a result, I have increased my score to a weak accept.

**Key Questions For Authors:**

1. How does the choice for the labeling distribution $\nu$ affect the method in practice? For instance, if you chose a uniform distribution, or a normal with larger or smaller variance? Would different choices affect the memorization-generalization trade off?
2. In the empirical experiments (section 4 and section 5), do all examples assume paired trajectory data / pairs samples $(x_t, x_{t+1})$? I see in the limitations the authors discuss that strong couplings between states is required. Possibly stating this clearly earlier in the text would be helpful for the reader.
3. The claim on lines 351-355: “ thus deterministic approaches such as neural operators cannot capture the stochasticity induced by the random media and permeability fields during the autoregressive roll out”, feels inadequately supported. Is the autoregressive diffusion model not also a stochastic approach? Could the authors compare their method to a _deterministic_ approach, one based on neural operators, to back this claim?
4. Could the authors comment on hyperparameter selection for their proposed method and on how easy or difficult it is to train their model using their proposed method? How does optimization compare to baselines on the video generation task?


**Minor comments**:

- The header title is left as the default ICML template header and should be changed to the respective title of the paper.

I am happy to adjust my score if the authors address my question and concerns through the rebuttal.

**Limitations:**

The authors provide a transparent discussion of limitations in their conclusion. The authors provide an impact statement after their conclusion.

**Strengths And Weaknesses:**

**Strengths**:
- This paper is written quite well, and relatively self-contained, barring some minor concerns that can be fixed quite easily (see first weakness below). The authors demonstrate a reasonably sound understanding of the related literature. The figures are clear, easy to follow, with informative captions.
- The authors provide theoretical support for their proposed methods, including error and generalization bounds, which strengthen the claims of their work.
- The authors provide a thorough set of empirical experiments with comparison with many relevant baselines to evaluate their proposed approach and support their claims.

**Weaknesses**:
- Paper organization could be improved. For example, Section 5 (Numerical Experiments), the authors conduct several different empirical experiments, but the section is split using bolded paragraph sentences. I recommend the authors use sub-section to better organize this section. I find at times section 5 was hard to follow with the current organization.
- This is a minor point, since the authors include a fairly comprehensive literature review in section 2, but there are no citations/references to literature in the introduction. In fact, the first citation does not appear till page 3. There are many statements in the introduction that warrant citations to literature, to name a couple: “general-purpose generative models such as diffusion- and flow-based approaches can represent distributions, but they typically require multi-step sampling at inference” and “recent work on one-step samplers aim to reduce … “. I encourage the authors to add references where relevant in the introduction.
- Information regarding how baselines are trained and optimized seems missing, making it difficult to assess the fairness of the comparisons.

---

> ### Author Rebuttal · Authors · 2026-03-30
>
> We thank the reviewer for the careful and constructive review. We appreciate the recognition of the theoretical support, thorough experiments, and the clarity of our figures. We will address the raised points below.
>
> ---
>
> ***"Paper organization could be improved"***
>
> - We agree and will restructure Section 5 into clearly labeled subsections (5.1 Wave & Flow, 5.2 CLEVRER, 5.3 BAIR) in the revision. We will also fix the default ICML header.
>
> ***"There are no citations/references to literature in the introduction … many statements in the introduction that warrant citations"***
> - Thank you for this; we will add citations for diffusion/flow-based models and one-step samplers directly in the introduction in the revision.
>
> ***"Information regarding how baselines are trained and optimized seems missing, making it difficult to assess the fairness of the comparisons."***
> - We will add a dedicated "Baseline Details" paragraph. For the ARDM baseline, we follow explicitly the set up of [Kohl et al., 2023] attempting to use all hyperparameters choices given in that paper as it is a standard reference for ARDMs applied to physics data. Importantly we specifically utilize the same UNet backbone and comparable parameter count as Stochastic Lifting. Otherwise, again following  [Kohl et al., 2023], the ARDM is trained with the same optimizer (Adam, β₁ = 0.9, β₂ = 0.999) and learning rate schedule (cosine lr = 1 × 10⁻⁴). The model conditions on 3 previous frames added along the channel dimension, uses a standard linear DDPM variance schedule.
> - We will include a table summarizing architecture, parameter count, optimizer, and training budget for every method.
>
> ***"How does the choice for the labeling distribution affect the method in practice? … uniform distribution, or a normal with larger or smaller variance?"***
>
> - We note that the key requirement from Proposition 3.3 is near-orthogonality of labels in high dimensions, which holds for any isotropic distribution with independent components. Numerical experiments verify this. On the BAIR benchmark we train with a uniform distribution and achieve an FVD of 71.2, nearly the same as with gaussian distribution.
> -  In terms of variance, it is standard to require network input to be at unit scale. It is well known that performance can degrade if this is not adhered to.
>
> ***"Do all examples assume paired trajectory data / pairs samples? … Possibly stating this clearly earlier in the text would be helpful"***
> - Yes, all experiments use paired trajectory data $(x_t, x_{t+1})$. We agree this should be stated upfront and will add an explicit remark in Section 1 (after introducing the transition law) clarifying that Stochastic Lifting requires paired sequential observations, not unpaired marginal samples.
>
> ***"Could the authors compare their method to a deterministic approach, one based on neural operators, to back this claim?"***
> - We have now run exactly this comparison. On the wave and flow benchmarks, we trained two deterministic baselines with MSE loss: (i) a UNet Operator with the identical architecture, parameter count, and optimizer as Stochastic Lifting but with labels removed, and (ii) a standard Fourier Neural Operator (FNO). Both are rolled out autoregressively at test time.
> - The both models perform very poorly on the distributional metrics we consider because they cannot produce variation across multiple rollouts. Thus the W_2 is measured between the true distribution and a dirac distribution.
> - This confirms the claim in Section 2.2: without an explicit source of stochasticity, even expressive neural operators collapse to mean behavior and cannot recover distributional structure.
> - Additionally we evaluate both models on the BAIR video benchmark. We note in this case FVD is also a metric which takes into account distributional characteristics and the ability to generate diverse rollouts. the FNO models perform very poorly achieving an FVD of 142.0 and the UNet archives a FVD of 96.0. Both of these are significantly worse than our Stochastic Lifting which gets an FVD of 69.0
>
>
> ***"Could the authors comment on hyperparameter selection … how easy or difficult it is to train … "***
>
> - Beyond standard architecture and optimizer choices (shared with all baselines), the only method-specific hyperparameter is the label dimension d, which is robust once sufficiently high (Figure 6). Training uses a standard regression loss with Adam and weight decay i.e.no auxiliary losses or scheduling tricks. For video, we only swap L2 for LPIPS.
> - This contrasts with ARDMs, which require tuning the number of diffusion steps s, the noise schedule, the sampling strategy, and optionally guidance scale and EMA decay; all of which interact and significantly affect quality, as reflected by the performance variation across step counts in Table 1. We will include a complete hyperparameter table in the revised manuscript.

---

> > ### Author Rebuttal · Reviewer_KT84 · 2026-04-03
> >
> > I thank the authors for the response to my questions and comments, and I will adjust my score accordingly.

---

> > > ### Author Response · Authors · 2026-04-06
> > >
> > > We thank the reviewer for their thoughtful discussion, which helped improve the quality of the paper. We are glad we were able to fully resolve their concerns and we thank them for agreeing to raise their score.

---

### Decision · Program_Chairs · 2026-04-30

**Decision:**

Accept (regular)

**Comment:**

The submission proposes an algorithm for generating trajectories of stochastic physical systems by appending an independently drawn random label to each training pair and training on the resulting data.

Overall, the reviews assess this paper positively. Reviewers KT84 and q5by appreciate the clear presentation and acknowledge that the theoretical contribution strengthens the claims of this work. Reviewers op4e, q5by, and y3je appreciate the simplicity of the proposed algorithm. The concerns raised by the reviews mainly pertain to some omitted details (KT84), selection of baselines (q5by), and questions about the algorithm's limit behaviour (op4e), which have been resolved to the reviewers' satisfaction in the rebuttals.

That said, Reviewer y3je expressed concerns about a disconnect between theory and experiments, which was only partly resolved during the rebuttal; Reviewer y3je's main remaining worry was that the changes would be too significant for a revision without resubmission. While I agree with Reviewer y3je that the changes are significant, I believe they are still manageable without resubmitting the manuscript.

Therefore, I recommend accepting this work contingent on implementing all changes promised in the rebuttals. Since the positive reviews of this paper largely depend on the revisions claimed in the rebuttals, implementing them is especially important.